# The Extracellular Matrix Proteins Tenascin-C and Tenascin-R Retard Oligodendrocyte Precursor Maturation and Myelin Regeneration in a Cuprizone-Induced Long-Term Demyelination Animal Model

**DOI:** 10.3390/cells11111773

**Published:** 2022-05-28

**Authors:** Juliane Bauch, Andreas Faissner

**Affiliations:** Department of Cell Morphology and Molecular Neurobiology, Ruhr-University Bochum, Universitätsstr. 150, 44801 Bochum, Germany

**Keywords:** tenascin-C, tenascin-R, myelin, oligodendrocyte, myelin lesion, cuprizone

## Abstract

Oligodendrocytes are the myelinating cells of the central nervous system. The physiological importance of oligodendrocytes is highlighted by diseases such as multiple sclerosis, in which the myelin sheaths are degraded and the axonal signal transmission is compromised. In a healthy brain, spontaneous remyelination is rare, and newly formed myelin sheaths are thinner and shorter than the former ones. The myelination process requires the migration, proliferation, and differentiation of oligodendrocyte precursor cells (OPCs) and is influenced by proteins of the extracellular matrix (ECM), which consists of a network of glycoproteins and proteoglycans. In particular, the glycoprotein tenascin-C (Tnc) has an inhibitory effect on the differentiation of OPCs and the remyelination efficiency of oligodendrocytes. The structurally similar tenascin-R (Tnr) exerts an inhibitory influence on the formation of myelin membranes in vitro. When Tnc knockout oligodendrocytes were applied to an in vitro myelination assay using artificial fibers, a higher number of sheaths per single cell were obtained compared to the wild-type control. This effect was enhanced by adding brain-derived neurotrophic factor (BDNF) to the culture system. *Tnr^−/−^* oligodendrocytes behaved differently in that the number of formed sheaths per single cell was decreased, indicating that Tnr supports the differentiation of OPCs. In order to study the functions of tenascin proteins in vivo *Tnc^−/−^* and *Tnr^−/−^* mice were exposed to Cuprizone-induced demyelination for a period of 10 weeks. Both *Tnc^−/−^* and *Tnr^−/−^* mouse knockout lines displayed a significant increase in the regenerating myelin sheath thickness after Cuprizone treatment. Furthermore, in the absence of either tenascin, the number of OPCs was increased. These results suggest that the fine-tuning of myelin regeneration is regulated by the major tenascin proteins of the CNS.

## 1. Introduction

Oligodendrocytes are the myelinating cells in the central nervous system (CNS). It is known that the production of myelin is essential for the normal functioning of the vertebrate CNS [1]. The oligodendrocytes derive from oligodendrocyte precursor cells (OPCs). These have a bipolar morphology at the beginning of their development and become progressively more complex with ongoing differentiation [2,3]. Finally, one single mature oligodendrocyte forms myelin membranes which can wrap around up to 40 different axons [4]. Oligodendrocytes fulfill many tasks, including the isolation of axons for better signal transmission with a top speed of 200 m/s [5], or the energy supplement of axons and the storage of glycogen [6]. The importance of oligodendrocytes is highlighted by demyelinating diseases such as multiple sclerosis, polyneuropathy, or neuromyelitis optica, for example [7]. Multiple sclerosis (MS) is a chronic inflammatory and demyelinating disease that affects the CNS by demyelinating the axons [8,9]. MS belongs to the autoimmune diseases and is the most common autoimmune disease in the world [10]. More than 2.8 million people are affected worldwide, especially young adults between the ages of 20 and 40 [11,12]. Interestingly, women are affected two to three times more often than men. During the course of the disease, oligodendrocytes are attacked and myelin membranes are destroyed by T-lymphocytes and activated macrophages, which results in demyelination and finally in damage of axons [10]. Furthermore, it recently became clear that B-cells also play an important role in MS by activating T-cells [10]. In a healthy organism, spontaneous remyelination processes occur in the demyelinated CNS, however, these do not suffice for a complete regeneration [13]. With regard to the clinical picture of MS, we focused on toxicity-based demyelination, namely the cuprizone model [14]. Cuprizone is a toxic copper chelator and induces demyelination, especially in the corpus callosum and the hippocampus of the rodent CNS. After the withdrawal of cuprizone, remyelination can be observed [14,15,16]. In this way, the mechanisms of remyelination or factors relevant for a higher remyelination efficiency can be analyzed in more detail. Referring to oligodendrocyte development, glycoproteins of the extracellular matrix (ECM) have been described to alter the differentiation and migration of oligodendrocytes [17,18]. The ECM is a gel-like network which consists of glycoproteins and proteoglycans and modulates parameters such as proliferation, migration, differentiation, morphology, and survival of cells. The ECM is synthesized and released by individual cells and allows the cohesion of tissue and organs [19,20,21]. Earlier studies revealed that the glycoproteins tenascin-C (Tnc) and tenascin-R (Tnr) similarly interfered with the formation of myelin membranes [22]. In general, tenascins constitute a subgroup of glycoproteins of the ECM and in mammals the four members tenascin-C, tenascin-R, tenascin-X, and tenascin-W are known [23]. Tnc has a hexameric structure and is synthesized by neural stem cells and immature astrocytes at the early postnatal stages of the CNS [24,25]. Previous studies showed that the glycoprotein Tnc has an inhibitory effect on the migration and differentiation of OPCs [22,26]. Furthermore, it has been reported that Tnc is upregulated in MS plaques which are characterized by demyelination of axons, and might convey inhibitory influences on oligodendrocytes [27]. Different from Tnc, Tnr is expressed during the late postnatal development of the CNS by a subclass of neurons and mature oligodendrocytes. Tnr has a similar structure to Tnc and its monomers assemble to dimers and trimers [22,24,28,29]. Tnr promotes cell adhesion and cell differentiation of oligodendrocytes, but also blocks the migration of OPCs [30]. So far it is known that both Tnc and Tnr exert a negative influence on the formation of myelin membranes and act antagonistically on the differentiation of oligodendroglia. [22]. Recently, it could be documented that both tenascins interfere with remyelination processes using an ex vivo model of organotypic cerebellar slice cultures [31]. In the light of these observations, we tested the hypothesis that these tenascins interfere with remyelination in vivo.

## 2. Materials and Methods

All performed experiments conformed to the relevant regulatory standards.

### 2.1. Animals and Genotyping

This study was carried out in accordance with the European Community Council Directive of 22 September 2010 (2010/63/EU) for care of laboratory animals and after approval of the local ethics committee (Landesamt für Umweltschutz, Naturschutz und Verbraucherschutz, Nordrhein Westphalen, LANUV: AZ 84-02.04. 2014.A332). All efforts were made to reduce the number of animals for this study in the best possible way. Mice (Mus musculus) were housed with a constant 12-h light–dark cycle and access to food and water ad libitum. For de- and remyelination, studies using cuprizone homozygous *Tnc^−/−^* and *Tnr^−/−^* mice were compared to in-house bred SV129 wild-type mice (Charles River, Sulzfeld, Germany). All experiments and animal handling were approved by the LANUV, North Rhine-Westphalia, Germany (AZ 84-02.04-2014.A332) and conducted according to German animal protection laws. For the cultivation of primary OPC cultures, heterozygous *Tnc^+/−^* and *Tnr^+/−^* mice were mated and the outcoming litters were genotyped after preparation. On this account, it was guaranteed that the cells used for the in vitro experiments had the same genetic background. In favor of the isolation of genomic DNA, mouse tail tips were lysed in 200 µL DirectPCR^®^ Tail Lysis Reagent (Cat. No.: 31-101-T, Peqlab, VWR Life Science; Radnor, PA, USA) containing 0.2 mg/mL Proteinase K (Cat. No.: EO0491, Thermo Fisher Scientific, Waltham, MA, USA) at 55 °C overnight. Then the tubes were incubated at 85 °C for 30 min and centrifuged at 16,000× *g* for 10 s. Therefore 0.5 µL of lysed genomic DNA was used as template for PCR analysis. The Tnr wild-type could be amplified by a Tnr primer (5′-AACTCCATGCTGGCTACCAC-3′) and Tnr Exon 1 primer (5′-TTTT GGGGAGGTT GATCTTG-3′), resulting in a PCR product of 420 bp. The amplification of the mutant allele with the Tnr primer combined with a Tnr Neo primer (5′-ACCGCTTCCTCGTGCTT-3′) resulted in a PCR product of approximately 429 bp. Tnr wildtype mice will be referred to as *Tnr^+/+^*, whereas Tnr knockout will be referred to as *Tnr^−/−^*.

### 2.2. Immunological Reagents

The following primary antibodies were used: APC (clone CC1, mouse IgG2b, IF: 1:100, Cat. No.: Ab16794, Abcam, Cambridge, UK), GFAP (1:300, DAKO, Cat. No.: Z0334; AB_10013382), Iba1 (019–19741; Wako and ab5076; Abcam), MBP (mouse IgG, IF: 1:50, Bio-Rad (MCA409) RRID: AB325004, Millipore, Burlington, MA, USA), Olig2 (rabbit polyclonal IgG, IF: 1:500, Merck (AB9610) RRID: AB_570666, Millipore) and antibodies against platelet-derived growth factor alpha receptor (PDGFRα) (rabbit polyclonal IgG, IF 1:300, Cat. No.: AB2174987, Santa Cruz, CA, USA). Species-specific secondary antibodies coupled to Cy2 AF488 (1:250) and Cy3 (1:500) were all obtained from Dianova GmbH (Hamburg, Germany).

### 2.3. Isolation and Cultivation of Primary Murine OPCs

As described in previous studies with minor modifications murine OPCs were isolated by immunopanning with a CD140a antibody [32,33]. First, the postnatal P6–P9 brains from the different genotypes *Tnc^−/−^*, *Tnc^+/+^*, *Tnr^−/−^* and *Tnr^+/+^* obtained by breeding of heterozygote mice were dissected. Then, the meninges were removed and enzymatically dissected in MEM containing 30 U/mL Papain (Cat. No.: LS003126, Worthington, Columbia, NJ, USA), 0.24 mg/mL L-cysteine (Cat. No.: C2529, Sigma-Aldrich, St. Louis, MO, USA) and 40 µg/mL DNAse I (Cat. No.: DNASE70, Worthington,) for 1 h at 37 °C. By the addition of the same amount of ovomucoid, the digestion was stopped. The cells were resuspended in panning buffer (PBS, 200 µg/mL (*w/v*) BSA fraction V (Cat. No.: A4919, Sigma-Aldrich), 5 µg/mL insulin (Cat. No.: 91077C, Sigma-Aldrich) and transferred to a culture dish pre-coated with anti-BSL1 Griffonia simplicifolia lectin (Cat. No.: B-1105, Vector Laboratories, Biozol, Eching, Germany), incubating 15 min at RT as negative selection. Then, the cells were relocated to a second dish pre-coated with AffiniPure goat anti-rat IgG (H + L) (Cat. No.: ab7481, Abcam) as a secondary antibody and rat anti-mouse CD140a (Biolegend, San Diego, CA, USA) as a primary antibody for positive selection. After 45 min of incubation culture dishes were washed with DMEM (Life Technologies by Thermo Fisher Scientific) and bound OPCs were detached with cell scrapers. In the following OPCs were cultivated in OPC SATO Medium DMEM (Cat. No.:41966029, Thermo Fisher Scientific, Waltham, MA, USA), Pen/Strep (Cat. No.: P4333, Sigma-Aldrich), 50 µg/mL (*w/v*) N-acetyl-cysteine, 10 ng/mL D-biotin (Cat. No.: B4639, Sigma-Aldrich), 5 µg/mL insulin (Cat. No.: I9278-5ML, Sigma-Aldrich), 2 µg/mL forskolin (Cat. No.: F6886, Sigma-Aldrich) and modified SATO 100 µg/mL apo-transferrin (Cat. No.: ab198654, Abcam), 100 µg/mL BSA (Cat. No.:8076.2, Carl Roth GmbH & Co. KG, Karlsruhe, Germany), 6.25 µg/mL progesterone (Cat. No.; P8783, Sigma-Aldrich), 16 µg/mL putrescine (Cat. No.: P5780, Sigma-Aldrich), 4 µg/mL sodium selenite (Cat. No.: 10102-18-8, Sigma-Aldrich) in the presence of 10 ng/mL platelet-derived growth factor AA (PDGF-AA) (Cat. No.: 100-13A, Peprotech, Waltham, MA, USA) and 5 ng/mL NT3 (Cat. No.: 450-03, Peprotech) in T-75 flasks (Cat. No.: 83.3911.002, Sarstedt, Nümbrecht, Germany) precoated with 10 µg/mL poly-D-lysine (PDL, Cat. No.: P0899, Sigma-Aldrich) at 37 °C and 7.5% CO_2_. After 1 week of cultivation, a sufficient number of OPCs proliferated and were ready for further experiments. During this period media were changed every 2–3 days and PDGF-AA was added daily. To detach the OPCs the dishes were washed once with PBS, then 5 mL of 0.25% Trypsin/EDTA (Thermo Fisher Scientific Inc., Cat. No.: 25300054) was added to OPCs for trypsinization. This process was stopped by the addition of an equal amount of ovomucoid (Leibovitz’s L15 medium (Cat. No.: L5520, Sigma-Aldrich), 1 mg/mL soybean trypsin inhibitor (Cat. No.: T6522, Sigma-Aldrich), 50 µg/mL BSA fraction V, 40 µg/mL DNAse I (Cat. No.: LS0020007, Worthington Biochem. Corp., Lakewood, NJ, USA)). After centrifugation for 5 min at 1000 rpm, the cell pellet was resuspended in OPC SATO medium and plated at a density of 20,000 cells per 12 mm coverslip. These were coated with 10 µg/mL (*w/v*) PDL beforehand. In the following cells were cultivated in the presence of 10 ng/mL PDGF-AA (Peprotech GmbH, Cat. No.: 100-13A) and 5 ng/mL NT3 (Cat. No.: 450-03, Peprotech) to maintain proliferation conditions. Alternatively, 400 ng/mL T3 (Cat. No.: T6397, Sigma-Aldrich) and 0.5% (*v/v*) horse serum (HS, Cat. No.: S9135, Peprotech) were added to start differentiation. The cells could be used for cultivation jointly with artificial fibers for 14 div and finally characterized by immunocytochemistry. 

### 2.4. Immunocytochemistry

Cells were stained immunocytochemically as previously described [34,35]. Medium from primary oligodendrocytes was removed and then the cells were washed twice with PBS and subsequently fixed with 4% (*w/v*) paraformaldehyde (PFA) (Cat. No.:4235.1, Carl Roth) in PBS for 10 min at RT. Thereafter, fixed oligodendrocytes were washed three times with PBT1 (PBS, 1% (*w/v*) BSA, 0.1% (*v/v*) Triton-X 100 (Cat. No.: A4975, AppliChem GmbH, Darmstadt, Germany)) and primary antibodies (anti-MBP, anti-GFAP) were diluted in PBT1 and incubated with the cells for 1 h at RT. Ensuing, oligodendrocytes were washed thrice with PBS/A before species-specific antibodies coupled to AF488, Cy2, or Cy3, and the nuclear marker Hoechst, which were diluted in PBS/A, were added to the oligodendrocytes for at least 1 h at RT. In the end, three washing steps in PBS were followed before the cells were mounted with Fluoromount-G (Cat. No.: 0100-01, Southern Biotech, Birmingham, AL, USA) and subjected to microscopy.

### 2.5. Myelination of Electrospun Fibers

For the analysis of the impact of the EMC molecules Tnc and Tnr on myelination in vitro, *Tnc^+/+^*, *Tnc^−/−^, Tnr^+/+^*, *Tnr^−/−^* and wildtype OPCs were plated onto parallelly aligned, 2 µm thick electrospun fibers. Those fibers are composed of poly-L-lactic acid (The Electrospinning Company, Didcot, UK) and presented in 12 well dishes [36]. First, the fibers were soaked in 70% (*v/v*) ethanol, followed by a coating with 10 µg/mL PDL for 1 h at 37 °C. After coating, the fibers were washed three times with H_2_O to remove the excess PDL. Then, OPCs were plated onto the fibers in a density of 45,000 cells per well in myelination medium (DMEM (Cat. No.: 41966029, Thermo Fisher Scientific Inc.): Neurobasal (Cat. No.: 21103-049, Sigma-Aldrich) (1:1), Pen/Strep, B27 (Cat. No.: 17504044, Thermo Fisher Scientific), ITS supplement (Cat. No.: I3146, Sigma-Aldrich), 5 μg/mL N-acetyl-cysteine (Cat. No.: A8199, Sigma-Aldrich), 10 ng/mL D-biotin (Cat. No.: B4639, Sigma-Aldrich), and modified SATO (100 μg/mL BSA fraction V, 60 ng/mL progesterone, 16 μg/mL putrescine, 400 ng/mL tri-iodothyronine [T3]; 400 ng/mL L-thyroxine [T4]) and cultivated at 37 °C and 7.5% CO_2_ for 14 days. Medium was changed three times per week. For immunocytochemical analysis, fibers were washed once in PBS and then fixed in 4% (*w/v*) PFA (Cat. No.:4235.1, Carl Roth) in PBS for 15 min at RT, followed by three washes in PBS. Additionally, the cells were permeabilized with 0.1% (*v/v*) Triton-X 100 in PBS for 15 min at RT. The primary antibodies were diluted in PBS and incubated at 4 °C overnight. On the next day, the primary antibody was removed, and fibers were washed three times with PBS. Next, the secondary antibodies were also diluted in PBS and added to the cells for 1 h at RT. After incubation, three washes with PBS followed. Last, fibers were mounted between glass slides and coverslips with Fluoromount-G (Cat. No.: 0100-01, Southern Biotech). Finally, the confocal images were obtained on a Zeiss LSM 510 Meta with an X40 oil objective. As many as necessary Z-steps of 0.37 μm were taken to image complete myelin sheaths. Analysis of the sheath lengths was determined with ImageJ by measuring the length of continuous membranes wrapping in a tube-like structure around the fibers. The number of sheaths formed by a single OL was also determined.

### 2.6. Cuprizone Model Induced Demyelination

In contrast to other models, the cuprizone model is a suitable model to study demyelination as well as remyelination [37]. For the analysis of demyelination and remyelination studies, 8-week-old male 129/SV, *Tnc^−/−^* and *Tnr^−/−^* mice were fed with 0.2% (*w/w*) cuprizone (Cat. No.: 370-81-0, Sigma-Aldrich) mixed with powdered chow for 10 weeks to induce demyelination [38], followed by a diet without cuprizone [33] for 2, 4, and 6 weeks to allow for a short- and long-term remyelination. Typically, mice are treated for 10 until 12 weeks to induce demyelination in a chronic way [39,40,41]. However, it could also be shown that after a 12-week cuprizone treatment remyelination is very sparse, resulting in a model of chronic demyelination [42] and remyelination that in some cases is insufficient, or even fails [43,44,45]. To ensure successful remyelination, we therefore decided to treat the mice for 10 weeks with cuprizone to mimic a more chronic course of demyelination. All in all, 12 mice per genotype were used for each condition, so that in total 180 mice were included in this study. However, not all individual animals were analyzed. During demyelination control mice received a normal diet of powdered chow (Appendix A), once a week cuprizone treated mice also received a normal diet to minimize the severity of intoxication. The weight of all animals was recorded three times a week. Here, one can see that the *Tnc^−/−^* mice have the least weight loss, yet also lose a lot of weight as demyelination increases. At the end of the experiment mice of each group were perfused intracardially with 20 mL PBS under deep anesthesia (800 µL 0.9% (*w/v*) NaCl (Cat. No.: 7647-14-5, Thermo Fisher Scientific), 50 µL Xylazin (10 mg/mL weight) (Cat. No.: 1205, CP-Pharma, Handelsgesellschaft GmbH, Burgdorf, Germany) and 150 µL Ketamin (150 mg/mL weight) (Cat. No.: 1202, CP-Pharma, Handelsgesellschaft GmbH). Brains were removed and cut sagittally in two halves, or the whole brain was used for electron microscopy analysis. For immunohistochemical stainings, one hemisphere was fixed with 4% (*w/v*) PFA (Cat. No.:4235.1, Carl Roth) in PBS at 4 °C for 48 h before embedding in tissue-freezing medium for cryosectioning. The second hemisphere was frozen in liquid nitrogen for RNA analysis, in particular for RT-PCR.

### 2.7. Histochemistry of the Brain Slices

After the two-day fixation in 4% (*w/v*) PFA (Cat. No.:4235.1, Carl Roth), the hemispheres were dehydrated in 30% (*w/v*) sucrose (Cat. No.: 4072-01, Fisher Scientific, Waltham, MA, USA) before embedding in tissue freezing medium (Cat. No.: 14020108926, Leica Biosystems, Nussloch, Germany) on dry ice. Afterwards, cryosections of 14 μm were cut from one hemisphere on a cryostat and stored at −20 °C. Here, the area on sagittal sections was focused on the corpus callosum (CC) above the hippocampus (HC) (according to Bregma: lateral 0.32–0.48 mm), where myelination is highest [46]. In order to provide proof for the efficacy of the induced demyelination with cuprizone, the Luxol-Fast-Blue-Periodic Acid Schiff (LFB-PAS) staining was used. Luxol fast blue marked myelin in blue and Periodic Acid and Schiff reagent was applied to label the axons in red color. In addition, hematoxylin was used to stain the cell nuclei dark blue. For the Schiff reagent staining with LFB-PAS cryosections were dehydrated in an increasing alcohol series starting at 30% (*v/v*), up to 96% (*v/v*) and stained in 0.1% (*w/v*) LFB solution (Cat. No.: 1328-51-4, Thermo Fisher Scientific), in 96% (*v/v*) ethanol (Cat. No.: 64-17-5, Carl Roth) for 24 h at 60 °C. Afterwards cryosections were briefly rinsed in 96% (*v/v*) ethanol before they were washed for 30 s in 0.05% (*w/v*) Lithium-Carbonate (LiCO_3_, Cat. No.: 554-13-2, Sigma-Aldrich). A staining with 1% (*w/v*) periodic acid for 7 min and afterwards with Schiff’s reagent (Cat. No.: 1789, Carl Roth) for 20 min followed. After cryosections had been washed hematoxylin staining was performed for 2 min. Finally, dehydration was carried out in an increasing series of alcohols from 70% (*v/v*) up to 100% (*v/v*) and cryosections were immobilized using Euparal mounting medium (Cat. No.: 7356.1, Carl Roth). For each condition and animal at least 2 sections were analyzed and at least 500 cells were examined.

For immunohistochemistry cryosections were first rehydrated in PBS before they were boiled in 0.01M citrate buffer for 70 min by 70 °C. Then the incubation with blocking solution (PBS, 1% (*w/v*) BSA, 0.1% (*v/v*) Triton-X 100, 5% (*v/v*) goat serum (Cat. No.: 005-000-121, Dianova GmbH) for 1 h at RT in a humid chamber followed. Primary antibodies were diluted in a blocking solution with goat serum and incubated at 4 °C overnight. After three consecutive washing steps in PBS, secondary antibodies were diluted in PBS/A (PBS, 0.1% (*w/v*) BSA) and incubated for 2 h at RT. Finally, stained cryosections were washed three times with PBS and mounted using ImmuMount (Cat. No.: 9990402, Thermo Fisher Scientific). The fluorescence stainings were recorded using the Axio Zoom.V16 (Cat. No.: 495010-0001-000, Carl Zeiss AG, Wetzlar, Germany), with a focus on the area of the corpus callosum. Here, the caudal part as well as the rostral part of the corpus callosum was analyzed in more detail, because these are the most affected areas. The corpus callosum is known to be subject to severe demyelination [15,47].

### 2.8. Molecular Biology

#### RNA Isolation, cDNA Synthesis, Polymerase Chain Reaction (PCR)

In order to monitor myelin-specific genes in the corpus callosum corresponding tissue was isolated from a half-frozen state of brain halves of mice exposed to cuprizone. Immediately after perfusion the isolated corpus callosum was stored at −80 °C until RNA was isolated. Total RNA from corpus callosum was obtained using the GeneElute™ Mammalian Total RNA MiniPrep Kit (Cat. No.: RTN350, Sigma-Aldrich) according to the manufacturer’s instructions. For analysis 0.5 µg RNA was transcribed into cDNA in a volume of 40 µL using the First strand cDNA synthesis Kit (Cat. No.: K1622, Thermo Fisher Scientific). For each condition (10 weeks control, 10 weeks demyelination, 2 weeks remyelination, 4 weeks remyelination, 6 weeks remyelination) cDNA samples from three different animals were prepared. We performed RT-PCR for the analysis of several genes which are relevant for oligodendrocyte development. In all conditions the housekeeping gene β-Actin was used as a control. Moreover, in each condition four animals with the following genes were investigated: platelet-derived growth factor receptor A (PDGFRα), myelin-basic protein (MBP), and ionized calcium-binding adapter molecule 1 (Iba1). Following primers were used: β-Actin forward: 5′-tatgccaacacagtgctgtctgg-3′, β-Actin reverse: 5′-agaagcacttgcggtgcacgatg-3′, PDGFRα forward: 5′-gcaccaagtcaggtcccatt-3′, PDGFRα reverse: 5′-cttcactggtggcatggtca-3′, MBP forward: 5′-tctcagccctgacttgttcc-3′, MBP reverse: 5′-atcaaccatcacctgccttc-3′, Iba1 forward: 5′-ggatttgcagggaggaaaag-3′, Iba1 reverse: 5′-tgggatcatcgaggaattg-3′. All RT-PCR results were analyzed with the rectangle tool ImageJ. In this context, the mean gray value of each sample was measured, the background was subtracted, and the resulting values were set in relation to the actin signal.

### 2.9. Electron Microscopy

Following the animal experimentation license the mice were anesthetized before they were first perfused intracardially with 10 mL PBS, followed by perfusion with 4% (*w/v*) PFA (Cat. No.: 4235.1, Carl Roth) and 2.5% (*v/v*) glutaraldehyde (Cat. No.: 111-30-8, Sigma-Aldrich) in 0.1 M phosphate buffer (pH = 7.4). The Brains were removed and kept in fixative for further 4 days. Subsequently, brains were cut in the coronal plane into 1 mm thick slices and the corpora callosa were dissected between Bregma −2.12 and 1.28 [33,48].

For each condition at least 8 sections were collected. Next the dissected corpora callosa were embedded in glycidether 100 (Cat. No.: 90529-77-4, Carl Roth). In the following ultrathin sections of the corpora, callosa were prepared, before they were stained with 2% (*w/v*) uranyl acetate (Cat. No.: 77870.02, Serva Electrophoresis GmbH, Heidelberg, Germany) for 5 min at RT and lead citrate (1.33 g Pb(NO_3_)2, 1.76 g Na3(C_6_H_5_O_7_) × 2 H_2_O, ad. 50 mL A. bidest) [49]. Finally, the sections were acquired on an electron microscope Sigma VP500 (Carl Zeiss AG, Wetzlar, Germany). The g-ratio (axon diameter divided by the total axon diameter including the myelin sheath) was determined. At least 200 axons for each condition per individual animal were evaluated. Overall, three animals for each condition (10 weeks control, 10 weeks demyelination, 10 weeks demyelination with 2, 4, or 6 weeks of remyelination) were examined. In each condition, the g-ratios of the knockouts were compared to the wildtypes.

### 2.10. Statistics

In order to examine the effects of the individual tenascins on the respective conditions, we always compared the results of the knockout situations with those of the wild types. Statistical analyses were carried out using the GraphPad Prism 7 software (GraphPad Software, San Diego, CA, USA). All results are provided as Mean ± SEM if not declared otherwise. Furthermore, the type of statistical tests and the number of performed experiments are provided in the figure legends. The significances were determined using ONE-way ANOVA with subsequent Tukey’s multiple comparisons test. In pairwise comparisons also the unpaired two-tailed Student’s *t*-test was used. The tests used are indicated in the figure legends. All statistical differences were considered as significantly different when *p* ≤ 0.05, *p*-values are referred as * for *p* ≤ 0.05, ** for *p* ≤ 0.01 and *** *p* ≤ 0.001. 

## 3. Results

### 3.1. Tenascins Intervene in Myelination of Artificial Microfibers

The reformation of myelin sheaths after damage requires the recruitment of OPCs to lesions and their local differentiation [7]. The ability to remyelinate axons seems to be an intrinsic property of oligodendrocytes [36]. In order to test if the elimination of tenascin genes affects the myelination capacity, *Tnc^+/+^*, *Tnc^−/−^*, *Tnr^+/+,^* and *Tnr^−/−^* oligodendrocytes were tested in a fiber myelination assay (Figure 1A). To this end, oligodendrocytes were cultivated in the presence of artificial fibers, and the cultures were investigated by immunocytochemistry using antibodies against GFAP and MBP (Figure 1B). Myelination could be revealed in each genotype and condition. Both the average number of fibers myelinated by a single cell which identifies the amount of myelin formed, as well as the numbers of sheaths extended per single oligodendrocyte were analyzed (Figure 1C–F). By comparing the average number of fibers ensheathed by single cells from *Tnr^+/+^* and *Tnr^−/−^* mice, it became clear that Tnr seemed to exert no influence on the extent of myelination of artificial fibers (*Tnr^+/+^*: 3.9 ± 0.2, *Tnr^−/−^*: 4.6 ± 0.2, *Tnr^−/−^* + BDNF: 7.2 ± 0.2, *p* < 0.0001; *Tnr^−/−^* vs. *Tnr^−/−^* + BDNF: *p* < 0.0001). However, it could be shown that Tnr seemed to have a positive influence on the number of sheaths formed per single cell. In the absence of Tnr the number of sheaths per single cell was reduced in comparison to the wildtype (*Tnr^+/+^*: 9.7 ± 0.7, *Tnr^−/−^*: 7.8 ± 1, *p* = 0.0049). This could indicate that Tnr promotes the outgrowth of myelin lamellae, a support that was not available when *Tnr^−/−^* were tested. The structural homologue Tnc interferes with membrane extension by oligodendrocytes in vitro [22] and was probed for influence on myelin sheath formation in the fiber assay. When OPCs prepared from *Tnc^−/−^ or* wildtype mice were compared the number of myelinated fibers did not differ (Figure 1E) while the number of sheaths per single cell was increased (Figure 1F).

Brain-derived neurotrophic factor is known to promote myelin repair in vivo [50]. Here, we wanted to test if BDNF has an influence on the knockout conditions and if knockout effects can even be boosted. Therefore, we supplemented the co-cultures with 10 ng/mL BDNF. Under these conditions, the average number of individual myelinated fibers more than doubled (*Tnc^+/+^*: 5.4 ± 0.2, *Tnc^−/−^*: 5.5 ± 0.2, *Tnc^−/−^* + BDNF: 14.7 ± 0.7; *Tnc^−/^*^−^ vs. *Tnc^−/−^* + BDNF: *p* < 0.0001) (Figure 1D,F). When the number of sheaths per individual cell was considered, BDNF likewise caused a significant increase (*Tnc^+/+^*: 6.4 ± 0.4, *Tnc^−/−^*: 7.9 ± 0.5, *p* = 0.027; *Tnc^−/−^* + BDNF: 11.7 ± 0.1, *p* < 0.0001; *Tnc^−/−^* vs. *Tnc^−/−^* + BDNF: 0.0086) (Figure 1F). Summarizing these results, the average number of ensheathed artificial fibers per cell did not depend on the genotype. However, Tnr and Tnc displayed opposite effects with regard to the number of individual sheaths per oligodendrocyte formed, in that *Tnr^−/−^* displayed a reduced and *Tnc^−/−^* OPCs an increased propensity to extend myelin lamellae. This is in agreement with our earlier report that Tnc interferes with while Tnr promotes OPC differentiation in culture [22]. The myelination capacity of OPCs could be strongly boosted by the addition of BDNF.

### 3.2. Electron Microscopy Analysis Revealed That Remyelination Is Accelerated in the Absence of Tenascins in the Cuprizone Animal Model

The result of the in vitro fiber myelination assay suggested that tenascin genes could intervene in remyelination after lesion. To test this aspect, both wildtype and knockout mice were subjected to cuprizone-mediated demyelination for a period of 10 weeks, modeling chronic demyelination [14,15]. The cuprizone model places the focus on the remyelination process in the absence of activated T-cells. Along this path we intended to analyze a more severe demyelination followed by a longer course of remyelination so that the influence of both tenascins on remyelination efficiency in what appears closer to a chronic setting could also be determined. Selected tissue arrays of the corpus callosum were analyzed at high resolution. To determine whether the cuprizone-induced demyelination as well as the remyelination efficiency after cuprizone withdrawal are working we performed LFB-PAS staining. As expected, the corpora callosa were stained in blue in the untreated control condition and the staining appeared weaker upon demyelination (Figure 2). Electron microscopy analysis revealed that the cuprizone-mediated demyelination was successful (Figure 3). The g-ratio, that is the ratio of the axon diameter divided by the diameter of the axon including the myelin sheath was plotted for the different experimental conditions (control, demyelination, remyelination 2, 4, and 6 weeks) using 129/SV wildtype and *Tnr* and *Tnc* knockout mice. The quantitative analysis of the three genotypes in each condition confirmed the impression that myelin sheaths were thinner in SV129 wildtype mice. Under control conditions (Figure 3a–c *Tnc^−/−^* and *Tnr^−/−−^* mice displayed thicker myelin membranes, as reflected by a significantly lower g-ratio (g-ratio C: 129/SV: 0.787 ± 0.004; *Tnc^−/−^*: 0.707 ± 0.005, *p* < 0.0001; *Tnr^−/−^*: 0.766 ± 0.004, *p* = 0.0001; *Tnc^−/−^* vs. *Tnr^−/−^*: *p* = 0.0344). The cuprizone treatment was effective, because upon demyelination the myelin sheaths appeared strongly reduced (Appendix A), as could be ascertained using histochemical LFB-PAS staining of sections (Figure 2). Disrupted and degraded myelin sheaths were visible in the treated tissues (Figure 3). Still, also upon demyelination both tenascin knockout lines revealed significantly thicker myelin sheaths than 129/SV mice (g-ratio D: 129/SV: 0.900 ± 0.002; *Tnc^−/−^*: 0.783 ± 0.005, *p* < 0.0001; *Tnr^−/−^*: 0.829 ± 0.005, *p* < 0.0001; *Tnc^−/−^* vs. *Tnr^−/−^*: *p* < 0.0001). After the withdrawal of cuprizone, remyelination occurred. After 2 weeks of remyelination thicker myelin sheaths could be observed in both tenascin knockouts (g-ratio 2R: SV/129: 0.835 ± 0.004, *Tnc^−/−^*: 0.763 ± 0.005, *p* < 0.0001; *Tnr^−/−^*: 0.812 ± 0.004, *p* = 0.0007; *Tnc^−/−^* vs. *Tnr^−/−^*: *p* < 0.0001).

After 4 weeks of remyelination this effect was even more pronounced in both tenascin knockout lines (g-ratio 4R: SV/129: 0.826 ± 0.004, *Tnc^−/−^*: 0.785 ± 0.004, *p* < 0.0001; *Tnr^−/−^*: 0.788 ± 0.004, *p* < 0.0001). With increasing duration of remyelination, the myelin membranes progressively recovered (Figure 3g–o). Consistently the lowest g-ratios were measured in *Tnc^−/−^* and in *Tnr^−/−^* mice (g-ratio 6R: SV/129: 0.809 ± 0.004; *Tnc^−/−^*: 0.763 ± 0.004, *p* < 0.0001; *Tnr^−/−^*: 0.797 ± 0.004; *Tnc^−/−^* vs. *Tnr^−/−^*: *p* < 0.0001) (Figure 3t). By comparing the g-ratios of the several remyelination conditions to demyelination, it became clear that remyelination is successful as the g-ratios in all three remyelination conditions (RM2, RM4, and RM6) were significantly reduced in wildtype mice (Appendix A) and decreased with ongoing remyelination time (g-ratio 129/SV 2R vs. 6R: *p* < 0.0001; 4R vs. 6R: *p* = 0.0023). This effect could also be observed in the knockout mice (g-ratios *Tnc**^−/−^* DM vs. 2R: *p* = 0.004; DM vs. 6R: *p* = 0.0034; 2R vs. 4R: *p* = 0.0028; 4R vs. 6R: *p* = 0.0024 and for *Tnr**^−/−^* 2R vs. 4R: *p* < 0.0001; 2R vs. 6R: *p* = 0.0023) (Appendix A). However, in *Tnr**^−/−^* mice the effective remyelination only evolved at the 4th week with a significant difference (g-ratios *Tnr**^−/−^* D vs. 4R: *p* < 0.0001; D vs. 6R: *p* < 0.0001) (Appendix A). Taken together the analysis revealed that in each condition the g-ratios of both knockouts were lower compared to the wildtype condition. This suggested that the ablation of Tnc as well as Tnr favored remyelination, possibly revealing inhibitory effects of tenascin proteins on myelin repair. The analysis of the axon diameter (Figure 4A) showed that upon demyelination conditions significantly smaller axon diameters were observable in both knockout genotypes (axon diameter D; SV129: 0.8496 ± 0.008, *Tnc^−/−^*: 0.6105 ± 0.012, *p* < 0.0001; *Tnr^−/−^*: 0.5751 ±0.0113, *p* < 0.0001). Also, in the untreated control condition there were significant differences with regard to axon diameters (axon diameter C; SV/129: 0.6241 ± 0.01, *Tnc^−/−^*: 0.5935 ± 0.01, *p* = 0.0323; *Tnr^−/−^*: 0.8714 ± 0.007, *p* < 0.0001; *Tnc^−/−^* vs. *Tnr^−/−^*: *p* < 0.0001) (Figure 4A). Interestingly, these differences were not detected after 2 weeks of remyelination (axon diameter RM2; SV/129: 0.57 ± 0.01, *Tnc^−/−^*: 0.5 ± 0.01, *Tnr^−/−^*: 0.6 ± 0.014). However, after 4 weeks of remyelination significantly lower axon diameters were detectable in *Tnc^−/−^* mice (axon diameter RM4; SV/129: 0.6254 ± 0.011, *Tnc^−/−^*: 0.5562 ± 0.006, *p* < 0.0001; *Tnr^−/−^*: 0.63 ± 0.01; *Tnc^−/−^* vs. *Tnr^−/−^*: *p* < 0.0001). Finally, after 6 weeks of remyelination significantly lower axon diameters were observed in both tenascin knockouts (axon diameter RM6; SV/129: 0.7631 ± 0.017, *Tnc^−/−^*: 0.6077 ± 0.01, *p* < 0.0001; *Tnr^−/−^*: 0.5728 ± 0.0105, *p* < 0.0001). Moreover, we plotted the g-ratios obtained against multiple axon diameters (Figure 4B–F). The linear regression slopes correlate axon diameter with g-ratios. The analysis of g-ratios that present the relation of myelin sheath thickness to axon diameter indicated differences between the mouse lines under study, both under control conditions and after demyelination (Figure 2). Although the lines have comparable SV129 backgrounds the independent breeding of the colonies may have resulted in small differences. Previous studies carefully worked out that both the *Tnc^−/−^* and the *Tnr^−/−^* knockout lines apparently do not display visible myelination deficits [51,52]. Thereby, an ascending slope indicates that the myelin sheath grows at a slower rate than the axon diameter while a descending slope reflects an increased growth of the myelin sheath with a growing axon diameter.

Concerning this parameter, our results confirmed that linear regression lines differed significantly, especially with regard to the *Tnr^−/−^* tissue (Figure 4B–F). Interestingly, after 4 weeks of remyelination the axons of the *Tnc^−/−^* and *Tnr^−/−^* mouse lines have acquired relatively larger myelin sheaths than the wildtype (Figure 4E). According to this parameter, the myelin sheaths in relation to the axon diameter were thinner in the wildtype than in the mutants also in the samples after demyelination and two weeks of remyelination (Figure 4C,D). The best fit lines which were obtained by using linear regression differed in each condition significantly between *Tnr^−/−^* and the 129/SV wildtype mice. However, in the untreated control condition, during demyelination, and in the early stage of remyelination the best fit lines between *Tnc^−/−^* and 129/SV wildtype mice were not significantly different (Figure 4B–D). Only after 4 weeks and 6 weeks of remyelination were the best fit lines between *Tnc^−/−^* and 129/SV wildtype mice significantly different (Figure 4E,F).

### 3.3. Tnc and Tnr Modulate Recruitment of OPCs to and Their Maturation in Myelin Lesions

To determine the role of both tenascins with regard to oligodendroglia recovery and myelin regeneration the markers Olig2 for the oligodendrocyte lineage in general and CC1 for differentiated oligodendrocytes were monitored by immunocytochemistry. Five experimental groups (10 weeks control (C), 10 weeks demyelination (DM), 2, 4, and 6 weeks of remyelination (RM)) including three different genotypes (SV/129, *Tnc^−/−^*, *Tnr^−/−^*) were analyzed (Figure 5). In the control group the expression of Olig2 in *Tnr^−/−^* mice was significantly reduced (C: SV/129: 84.77 ± 4.04%, C: *Tnc^−/−^*: 84.03 ± 6.46%, C: *Tnr^−/−^*: 65.29 ± 6.83%, *p* < 0.0001; *Tnc^−/−^* vs. *Tnr^−/−^*, *p* = 0.0002). This effect was also observable upon demyelination (Figure 5A(d–f),B), where also *Tnc^−/−^* possessed a lower number of Olig2 positive cells (DM: SV/129: 71.27 ± 7.77%, *Tnc^−/−^*: 60.45 ± 9.86%, *Tnr^−/−^*: 49.91 ± 10.71%, *p* = 0.0027) (Figure 5). Interestingly, in the early stage of recovery after 2 weeks of remyelination a significantly decreased number of Olig2 positive cells was detected in *Tnc^−/−^* (2RM: SV/129: 79.4 ± 7.2%, *Tnc^−/−^*: 41.2 ± 12.5%, *p* < 0.0001; *Tnr^−/−^*: 73.6 ± 6.5%; *Tnc^−/−^* vs. *Tnr^−/−^*, *p* < 0.0001) (Figure 5B). After 4 weeks of remyelination no significant differences between the genotypes were seen (4RM: SV/129: 74.4 ± 11.7%, *Tnc^−/−^*: 87.8 ± 2.6%, *Tnr^−/−^*: 79 ± 2.8%). This result persisted over a remyelination period of 6 weeks (6RM: SV/129: 83.3 ± 3.7%, *Tnc^−/−^*: 80.5 ± 2.4%, *Tnr^−/−^*: 90.2 ± 3%) (Figure 5B). As the maturation of OPCs is a prerequisite for myelin regeneration, we also determined the number of CC1-positive cells. In the untreated control it seems that more CC1 positive cells were present in *Tnc^−/−^* mice, however, this effect could not be statistically proven (C: SV/129: 55 ± 2.5%, *Tnc^−/−^*: 66.6 ± 6.2%, *Tnr^−/−^*: 51.9 ± 2.5%; *Tnc^−/−^* vs. *Tnr^−/−^*, *p* = 0.0352). During demyelination, the number of CC1 positive cells in each genotype was decreased in comparison to the untreated control condition. Initially, the genotypes did not differ (DM: SV/129: 40.6 ± 3.1%, *Tnc^−/−^*: 38.9% ± 3%, *Tnr^−/−^*: 47.3 ± 2.2%). However, in the early stage of recovery after 2 weeks of remyelination CC1-positive cells were significantly less in both *Tnc^−/−^* and *Tnr^−/−^* in comparison to SV/129 wildtype tissue (2RM: SV/129: 51.2 ± 3.6%, *Tnc^−/−^*: 32.3 ± 3%, *p* = 0.0024; *Tnr^−/−^*: 30.3 ± 2.6%, *p* = 0.0008) (Figure 5C). This effect vanished after further 2 weeks of remyelination, where the number of CC1-positive cells was equivalent in *Tnr^−/−^* and SV129 wildtype mice. Interestingly, at that stage *Tnc^−/−^* tissue displayed substantially more CC1-positive than the *Tnr^−/−^* and SV/129 wildtype (4RM: SV/129: 55.2 ± 3%, *Tnc^−/−^*: 79.3 ± 2.7%, *p* = 0.0001; *Tnr^−/−^*: 55.8 ± 1.1%; *Tnc^−/−^* vs. *Tnr^−/−^*, *p* < 0.0001). Finally, after 6 weeks of remyelination *Tnc^−/−^* and SV/129 mice were comparable, while the fraction of mature oligodendrocytes appeared elevated in *Tnr^−/−^* mice (6RM: SV/129: 65.4 ± 2.9%, *Tnc^−/−^*: 67.4 ± 2.6%, *Tnr^−/−^*: 88.02 ± 1.9%, *p* < 0.0001; *Tnc^−/−^* vs. *Tnr^−/−^*, *p* < 0.0001) (Figure 5C). The myelination process involves membrane extension, initial axonal contact, and subsequent stabilization [53]. Consequent to contact the proteolipid-protein 1 (PLP) and MBP mRNA are transported to the plasma membrane where the MBP synthesis is carried out [54,55]. The analysis of mRNA levels for MBP was performed to obtain more insight into oligodendrocyte maturation in our samples (Figure 5D). MBP represents a well-established biomarker for the maturation of myelin membranes [56,57]. There were no apparent differences between the genotypes of the untreated controls (C: SV129: 2 ± 0.4, *Tnc^−/−^*: 2.8 ± 0.4, *Tnr^−/−^*: 1.97 ± 0.4) and after demyelination (DM: SV129: 1.6 ± 0.1, *Tnc^−/−^*: 1.8 ± 0.2, *Tnr^−/−^*: 1.9 ± 0.2). Interestingly, in the early stage of recovery after 2 weeks of remyelination the MBP expression was significantly increased in *Tnc^−/−^* (*p* = 0.0342) as well as in *Tnr^−/−^* (*p* = 0.0187) in comparison to wild-type mice (2R: SV129: 1.9 ± 0.01, *Tnc^−/−^*: 2.4 ± 0.1, *p* = 0.0342; *Tnr^−/−^*: 2.4 ± 0.1, *p* = 0.0187). This might indicate that MBP was upregulated to compensate for the delay in maturation towards CC1-positive cells observed during that phase (Figure 5C). After 4 weeks of remyelination, no differences were visible (4RM: SV129: 2.1 ± 0.3, *Tnc^−/−^*: 2.4 ± 0.2, *Tnr^−/−^*: 2.5 ± 0.1). Thereafter, MBP message levels increased further in both tenascin knockouts, paralleling the advent of CC1-positive cells (6RM: SV129: 2.1 ± 0.1, *Tnc^−/−^*: 2.5 ± 0.1, *p* = 0.0135; *Tnr^−/−^*: 2.5 ± 0.1, *p* = 0.1311) (Figure 5C,D). The analysis of the oligodendrocyte population was complemented by an investigation of oligodendrocyte precursor cells using the marker platelet-derived growth factor alpha (PDGFRα), which is an established marker for OPCs (Figure 6). Under control conditions more PDGFRα-positive cells were visible (Figure 6A(a–c)) in both tenascin knockout lines (C: SV/129: 11.3 ± 1.5%, *Tnc^−/−^*: 20.2 ± 1.2%, *p* = 0.0019; *Tnr^−/−^*: 21.3 ± 2.6%, *p* = 0.0115).

These results could also be observed in demyelination condition, in which especially in *Tnc^−/−^* mice the number of OPCs increased substantially in comparison to the SV/129 wildtype and untreated condition. Furthermore, in *Tnr^−/−^* mice the number of OPCs was also significant increased (DM: SV/129: 9.8 ± 2%, *Tnc^−/−^*: 21.3 ± 2.6%, *p* = 0.0005; *Tnr^−/−^*: 19.4 ± 2%, *p* = 0.0091). The largest number of PDGFRα positive oligodendrocytes was detected in the early stage of recovery in *Tnc^−/−^* mice after 2 weeks of remyelination (Figure 6B), whereas there were no significant differences between the *Tnr^−/−^* and SV/129 measurable (2RM: SV/129: 20.4 ± 1.2, *Tnc^−/−^*: 39.5 ± 2%, *p* < 0.0001; *Tnr^−/−^*: 28.6 ± 0.8, *p* = 0.0004; *Tnc^−/−^* vs. *Tnr^−/−^*: *p* = 0.0117).

After 4 weeks of remyelination no differences between the several genotypes were detectable (4RM: SV/129: 23.4 ± 1.2%, *Tnc^−/−^*: 23.4 ± 1.5%, *Tnr^−/−^*: 24.2 ± 1%). These relations remained unchanged even after 6 weeks of remyelination (6RM: SV/129: 20.3 ± 2.1%, *Tnc^−/−^*: 22 ± 2.3%, *Tnr^−/−^*: 14.6 ± 3%). Taken together these results show that especially in the absence of Tnc significantly more OPCs were detectable. The analysis of the PDGFRα expression was also performed on the message level (Figure 6C). The determination of relative message levels indicated no differences between the genotypes in the control (C: SV129: 0.355 ± 0.017, *Tnc^−/−^*: 0.348 ± 0.023, *Tnr^−/−^*: 0.448 ± 0.09) and after demyelination (DM: SV129: 0.36 ± 0.05, *Tnc^−/−^:* 0.29 ± 0.05, *Tnr^−/−^*: 0.28 ± 0.013). The sole differences that emerged with regard to the message level were the stages of 2 weeks (2RM: SV129: 0.393 ± 0.033, *Tnc^−/−^*: 0.663 ± 0.12, *Tnr^−/−^*: 0.29 ± 0.024, *p* = 0.0413) and 4 weeks (4RM: SV129: 0.473 ± 0.036, *Tnc^−/−^*: 0.868 ± 0.114, *p* = 0.0161; *Tnr^−/−^*: 0.378 ± 0.041) of regeneration. Thus, after 4 weeks a clearly increased upregulation occurred in the *Tnc^−/−^* mutant (Figure 6C). Interestingly, this was also reflected in the number of PDGFRα-positive genotypes tested (6RM: SV129: 0.45 ± 0.04, *Tnc^−/−^*: 0.48 ± 0.05, *Tnr^−/−^*: 0.59 ± 0.10).

### 3.4. The Loss of Tnc Enhances Astrocyte Reactivity in Cuprizone-Induced CNS Lesions

The reactivity of astrocytes is a hallmark of tissue lesions in the CNS [58]. The distribution of astrocytes was determined by using GFAP as a marker (Figure 7). Under control conditions, less astrocytes were counted in both knockout lines (C: SV/129: 23.45 ± 2.3%, *Tnc^−/−^*: 7.8 ± 1.8%, *p* = 0.0016; *Tnr^−/−^*: 9.1 ± 1.4%, *p* = 0.0017^) (^Figure 7B). As the myelin was degraded by the cuprizone diet, the response to lesions was of particular interest. There, we observed a smaller fraction of GFAP-positive cells (Figure 7A(d–f),B) in the *Tnr^−/−^* after demyelination (DM: SV/129: 22.33 ± 3.3%, *Tnc^−/−^*: 33.8% ± 7.2%, *Tnr^−/−^*: 10.12 ± 1.2%, *p* = 0.0136; *Tnc^−/−^* vs. *Tnr^−/−^*: *p* = 0.0115). Within the first two weeks of remyelination, however, a strong increase of GFAP-positive cells could be noted in the *Tnc^−/−^* tissue, and a higher proportion of GFAP-positive cells was seen in the *Tnr^−/−^* (2RM: SV/129: 11.6% ± 1.9%, *Tnc^−/−^*: 38.1 ± 7.8%, *p* = 0.0165; *Tnr^−/−^*: 21.45 ± 2.8%, *p* = 0.0266) (Figure 7B). These differences were no longer visible after 4 weeks (4RM: SV/129: 17.6 ± 3.8%, *Tnc^−/−^*: 21.3 ± 1.6%, *Tnr^−/−^*: 8 ± 1%, *p* = 0.0508; *Tnc^−/−^* vs. *Tnr^−/−^*: *p* = 0.0110) and 6 weeks of regeneration (6RM: SV/129: 17.1% ± 3.6%, *Tnc^−/−^*: 14.8 ± 1.4%, *Tnr^−/−^*: 18.3 ± 1.6%) (Figure 7A(j–o) and Figure 2B). Thus, astrocyte reactivity was increased in both tenascin knockout lines within the first two weeks after the lesion.

### 3.5. Tenascins Modulate Microglia and Leucocytes in Cuprizone-Induced Lesions

Increasing evidence indicates that the ECM and tenascins regulate also the immune reactions upon lesion to the CNS [59,60]. Therefore, we investigated the local immune response including microglia and macrophages using the markers Iba1 and CD68 expression in our demyelination model (Figure 8). Interestingly, in the untreated condition the number of Iba1 positive cells appeared significantly increased in both tenascin knockouts (C: SV/129: 8.9 ± 0.8%, C: *Tnc^−/−^:* 15.3 ± 1%, *p* = 0.001; C: *Tnr^−/−^*: 11.74 ± 0.9%, *p* = 0.0443) (Figure 8B). It is known that microglia are repulsed by anti-adhesive Tnr in vitro [61]. Upon demyelination the number of Iba1-positive cells overall augmented, most extensively in the *Tnr^−/−^* (DM: SV/129: 22.04 ± 1.4%, *Tnc^−/−^*: 26.22 ± 1.7%, *Tnr^−/−^*: 30.16 ± 2.8%, *p* = 0.0329) (Figure 8B). In the first recovery phase, the microglia stayed elevated, though less so in the *Tnc^−/−^* tissue (2RM: SV/129: 24.62 ± 1.12%, *Tnc^−/−^*: 20.6 ± 1.03%, *p* = 0.0292; *Tnr^−/−^*: 27.38 ± 0.71%). Microglia populations were comparable after 4 weeks recovery (4RM: SV/129: 26.14 ± 2.8%, *Tnc^−/−^*: 22.58 ± 1.21%, *Tnr^−/−^*: 19.36 ± 1,08%) and decreased further after 6 weeks, with relatively higher proportion of microglia in the *Tnc^−/−^* (6RM: SV/129: 13.98 ± 0.93%, *Tnc^−/−^*: 17.76 ± 1.08%, *p* = 0.0295; *Tnr^−/−^*: 16.54 ± 1.85%) (Figure 8A(j–o),B). The microglia and macrophage compartment were investigated further using the marker CD68 that was evaluated by RT-PCR analysis (Figure 8C). CD68 is a highly glycosylated glycoprotein which is expressed by mononuclear phagocytes and used to detect the response of activated microglia as well as macrophages that accumulate in acute lesions [62]. In the untreated situation CD68 expression was higher in *Tnr^−/−^* mice (C: SV/129: 0.46 ± 0.06, *Tnc^−/−^*: 0.58 ± 0.12, *Tnr^−/−^*: 0.86 ± 0.14). Demyelination strongly boosted CD68 expression in all genotypes, most intensely so in the wildtype and the *Tnc^−/−^* mouse line (DM: SV/129: 2.24 ± 0.15, *Tnc^−/−^*: 2.28 ± 0.19, *Tnr^−/−^*: 1.16 ± 0.12, *p* = 0.0005; *Tnc^−/−^* vs. *Tnr^−/−^*: *p* = 0.0023). Thereafter, CD68 expression steadily decreased with ongoing recovery, although more rapidly in the *Tnr^−/−^* knockout after 2weeks (2RM: SV/129: 1.9 ± 0.18, *Tnc^−/−^*: 1.45 ± 0.23, *Tnr^−/−^*: 1.37 ± 0.1, *p* = 0.0288). After 4 weeks of recovery CD68 went further down in all genotypes (4RM: SV/129: 1.33 ± 0.19, *Tnc^−/−^*: 1.53 ± 0.3, *Tnr^−/−^*: 0.97 ± 0.11) and reached basal level in the wildtype after 6 weeks (6RM: SV/129: 0.61 ± 0.13, *Tnc^−/−^*: 2.1 ± 0.5, *p* = 0.0172; *Tnr^−/−^*: 1.02 ± 0.02, *p* = 0.0234). The knockout lines still showed enhanced activation with elevated CD68 at that stage. Clearly, the demyelination by cuprizone treatment led to an activation of the microglia compartment that appeared more accentuated in several instances in the absence of Tnc or Tnr.

## 4. Discussion

In the present study, we investigated the influence of the glycoproteins Tnc and the structurally related Tnr on the regenerative response of oligodendroglia in a myelin lesion paradigm using cuprizone. Previous studies in the laboratory had revealed that the Tnc protein of the tenascin family of ECM glycoproteins exerts inhibitory effects on the motility and differentiation of oligodendrocyte precursor cells in vitro [22,26,63]. Although both glycoproteins have a similar structure, their expression patterns differ. Tnc is expressed by neural stem and astrocyte precursor cells and down-regulated postnatally. Tnr is restricted to the CNS and expressed postnatally by maturing oligodendrocytes and a subpopulation of neurons [22,25,64]. By comparing *Tnc^+/+^* and *Tnc^−/−^* OPC cultures in the artificial fiber assay we obtained evidence that Tnc has a negative impact on the number of myelin sheaths per oligodendrocyte and thereby the extent of myelin formation. This is consistent with previous reports that showed an interference of Tnc with oligodendrocyte membrane extension [22]. Thus, as expected, *Tnc^−/−^* OPCs generated a higher number of myelin extensions per cell than the wildtype. This beneficial effect was strongly boosted by the addition of BDNF, a neurotrophin that is known to support myelin regeneration in vivo [50,65,66,67]. Inhibition of membrane extension may be rooted in the fact that Tnc prevents RhoA activation [68,69], because the reduction of RhoA activation by deletion of the nucleotide exchange factor Vav3 led to reduced myelination, also in vivo [70,71]. In the artificial fiber myelination assay *Tnr^−/−^* OPCs behaved differently in that the number of myelin extensions per cell was reduced. At first sight, this was unexpected because Tnr also suppresses RhoA activation and membrane formation, similar to Tnc [22].

However, both Tnc and Tnr have opposite effects on oligodendrocyte maturation in that Tnc prevents whereas Tnr promotes the maturation of OPCs towards the expression of MBP [22,30]. Thus, the tampered ability of *Tnr^−/−^* OPCs to extend membranes may result from a maturation deficit towards the myelinating stage, a process that conversely would benefit from a deficit of Tnc. Tnc is known to modulate the adhesion of various cell types to culture substrates [72], yet the *Tnc^−/−^* mouse lines are viable and fertile, in the absence of gross morphological or functional deficits [73], and in particular the myelin compartment appeared developmentally unperturbed [52]. On the other hand, several studies revealed subtle developmental modifications in the radial glia and oligodendrocyte compartments that were compensated for in the adult CNS [74,75]. Furthermore, recent studies suggested functional roles of Tnc in response to lesions, in particular in the CNS [76,77,78,79], reviewed in [80]. Therefore, we decided to investigate the roles of Tnc and Tnr in a myelin lesion model based on the application of cuprizone to the chow [81,82]. In this approach, the newly formed myelin sheaths are generally thinner and shorter than the former ones [48,83,84]. The effects on myelin formation were monitored by electron microscopy and by examining well-established markers of oligodendrocyte progenitor maturation and the major macroglial cell types. A strong demyelination could be achieved by dispensing a cuprizone-rich diet for 6 weeks [31,81]. In our study, this effect was clearly visible after 10 weeks of treatment, which we consider to mimic chronic demyelination. Thereafter, remyelination ensued as expected [85,86]. Currently, chronic demyelination is induced by a 12-week cuprizone treatment [40,41]. However, in these cases, the remyelination fails or is insufficient [43,44,45,87] due to the reduced expression of Trem2 resulting in an impaired clearance of myelin debris from microglia. Therefore, we opted for a 10-week cuprizone treatment to mimic a chronic course of demyelination. According to the g-ratio, remyelination was more efficient in the tenascin knockouts than in the wildtype, and most effective in the absence of Tnc. This does not exclude a differential sensitivity to Cuprizone treatment. Interestingly, known effects concern the roles of Tnc in regulating the proliferation and migration of OPCs [74]. Furthermore, an anti-apoptotic effect of Tnc for oligodendrocytes in vitro has been reported [63]. Less is known in this regard about Tnr. Therefore, the question remains open whether some of the differences in de- and remyelination observed are due to a differential susceptibility of the mouse lines towards Cuprizone treatment.

The phenotype in the *Tnc^−/−^*-line most probably reflects the inhibitory properties of Tnc for maturation and membrane extension observed in *vitro*. This effect of Tnc was accompanied by a strong upregulation of GFAP, reflecting enhanced astrocyte reactivity in this situation. Interestingly, a strong upregulation of GFAP had also been observed in a stab wound model to the cortex of the *Tnc^−/−^* mouse line [88]. Interestingly, our laboratory has shown that the proliferation of astrocytes is augmented in the embryonic *Tnc^−/−^* spinal cord, which may be responsible for the enhanced astroglial reactivity observed in lesions [89]. Reactive astrocytes can also be considered a source for the release of Tnc in lesioned tissue [90]. Of note, it has been proposed that astrocyte-derived Tnc contributes to the inhibitory environment for remyelination in Multiple Sclerosis lesions [91,92]. It has been shown that alphaV integrins are upregulated during remyelination, in conjunction with Tnc and Tnr that may represent functional ligands in that context [27]. Integrins are considered important receptors for tenascin functions [93,94]. Beyond its role in perineuronal nets where it serves as a major scaffolding protein [95], Tnr has also been studied in lesion situations. In this context, it has been proposed as an inhibitory axonal guidance molecule, notably in the optic nerve [29] and in the spinal cord [96]. As Tnr is expressed by adult oligodendrocytes it may be part of degraded myelin that abounds in demyelinating lesions. There, it also seems to interfere with remyelination, as the *Tnr^−/−^* mice showed an improved recovery in the cuprizone model, albeit not as effectively as the *Tnc^−/−^* counterpart. Both tenascins interact with various chondroitin sulfate proteoglycans (CSPGs) of the lectican family such as versican and brevican and thus partake in the inhibitory environment that restricts regeneration of myelin, in particular in multiple sclerosis lesions [97,98,99]. The copper chelator cuprizone sets a metabolic insult that preferentially leads to the elimination of mature oligodendrocytes by apoptosis [15,82]. The elimination of oligodendrocytes was particularly effective in the *Tnc^−/−^* tissue where the lowest number of Olig2-positive cells was recorded. This is in agreement with the former report that OPC proliferation is reduced in the Tnc knockout [74]. Repair of the deficit requires the recruitment of OPCs to the lesion site, where the cells have to differentiate towards the myelin-forming stage [7,8]. Tnc is known to interfere with OPC motility in vitro [26,100] and *Tnc^−/−^* knockout mice revealed an accelerated invasion of OPCs into the optic nerve [74]. Similarly, in our current study more PDGFR-positive OPCs [101,102] were detected upon demyelination and in the first two weeks of remyelination in the absence of Tnc. Tnr embodies anti-adhesive properties for different neural cell types [103]. While its impact on migration has not explicitly been studied this may explain the increased immigration of PDGFR-positive OPCs observed upon demyelination and during the first two weeks of remyelination, analogous to albeit not as extensive as in the *Tnc^−/−^* situation. Once on site, the OPCs have to engage in differentiation in order to remyelinate the axon. Myelin basic protein (MBP) is very important for the compaction of the myelin sheaths [56]. Tnc is known to delay the maturation of OPCs towards the MBP-expressing mature state [22]. This effect is mediated by the receptor contactin-1 (Cntn1), the activation of downstream signaling pathways, and the suppression of the RNA-binding protein Sam68, an oligodendrocyte maturation factor [34]. In agreement with these inhibitory properties, we observed an increase in mature CC1-positive oligodendrocytes and MBP expression within two to four weeks of remyelination in the *Tnc^−/−^* system. Tnr, different from Tnc promotes the acquisition of MBP and has been proposed as an autocrine maturation factor of oligodendrocytes in vitro [22,30]. Notwithstanding, the maturation of OPCs in the absence of Tnr was not compromised, which might indicate that the loss of Tnr was compensated by axon-derived signals in the in vivo situation.

It is interesting to compare our results with studies that focused on autoimmune processes in tenascin knockout mutants. Thus, in an experimental autoimmune encephalitis elicited by the injection of the myelin-oligodendrocyte glycoprotein (MOG), the encephalitogenic response of Th1 and Th17 immune cells was substantially reduced in the absence of Tnc [76]. Likewise, in an autoimmune glaucoma model, the ablation of Tnc resulted in amelioration of outcome [78]. These observations conform with a recent report that Tnc belongs to the damage associated molecular patterns (DAMPs) and is as efficient regarding the activation of microglia as lipopolysaccharides [104]. These findings stage Tnc as an important modulator of immune responses in the CNS [17,60]. For example, Tnc associated with exosomes has been found to suppress T-cell activation [105] and the extracellular matrix appears involved in the immune response to ischemia [59,106]. In our studies, we saw several differences between the genotypes regarding the number of Iba1-positive cells and the expression of CD68, that was particularly upregulated after six weeks of remyelination in the *Tnc^−/−^* lesions. Interestingly, a recent report emphasized a regulatory role of Tnc for microglia surveillance and leucocyte infiltration in ischemic CNS infarct territory [106].

Summarizing our findings, it seems that the elimination of Tnc and Tnr exerts a clear impact on the remyelination of cuprizone-induced myelin degradation. Both the recruitment of OPCs to the lesion territory and their local differentiation to myelinating cells are promoted in the absence of tenascins. Consequently, remyelination is supported and myelin repair is accelerated in the mutants.

## Figures and Tables

**Figure 1 cells-11-01773-f001:**
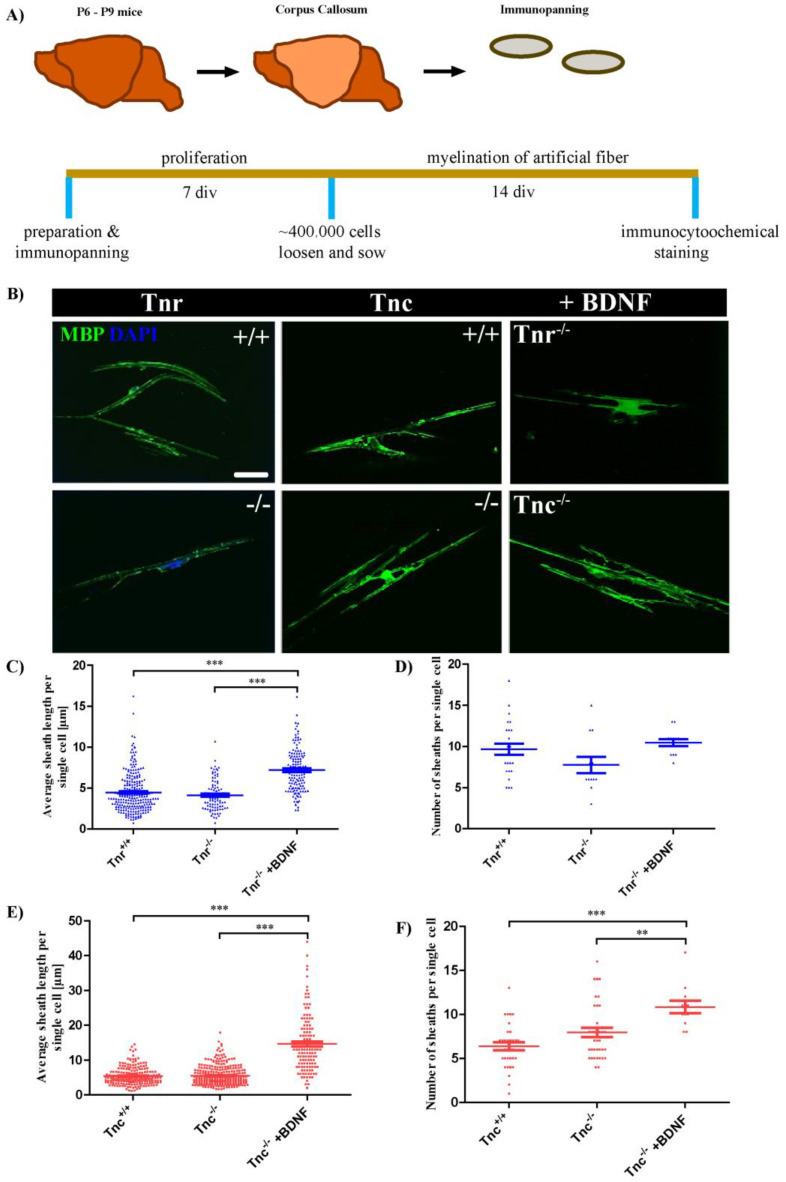
Tnc retards whereas Tnr and BDNF promote myelination by OPCs in an artificial fiber assay. (**A**) OPCs from P6–P9 mice from 129/SV wild-type, *Tnc^−/−^* and *Tnr^−/−^* mice were prepared via immunopanning and cultivated for 7 *div*. OPCs were seeded on artificial fibers and cultivated for 14 *div*. Exemplary photomicrographs of artificial poly-L-lactic acid electrospun microfibres with OPCs in myelination medium for 14 *div* are shown (**B**) Immunocytochemical staining was performed with antibodies against GFAP (to exclude astrocytes) and MBP (green, to determine myelin). Scale bar: 50 µm. To determine the myelination degree in the different conditions the average sheath length of fibers per single cell (**C**,**E**) and the number of sheaths per single cell (**D**,**F**) were determined. In a pilot study, BDNF was added to *Tnc^−/−^* and *Tnr^−/−^* OPCs to analyze its impact on myelination (**E**,**F**). A minimum of 35 single cells in 3 independent experiments were analyzed (n = 35, N = 3). For statistical analysis, the unpaired two-tailed student’s test was used (*p* ≤ 0.01 **, *p* ≤ 0.001 ***).

**Figure 2 cells-11-01773-f002:**
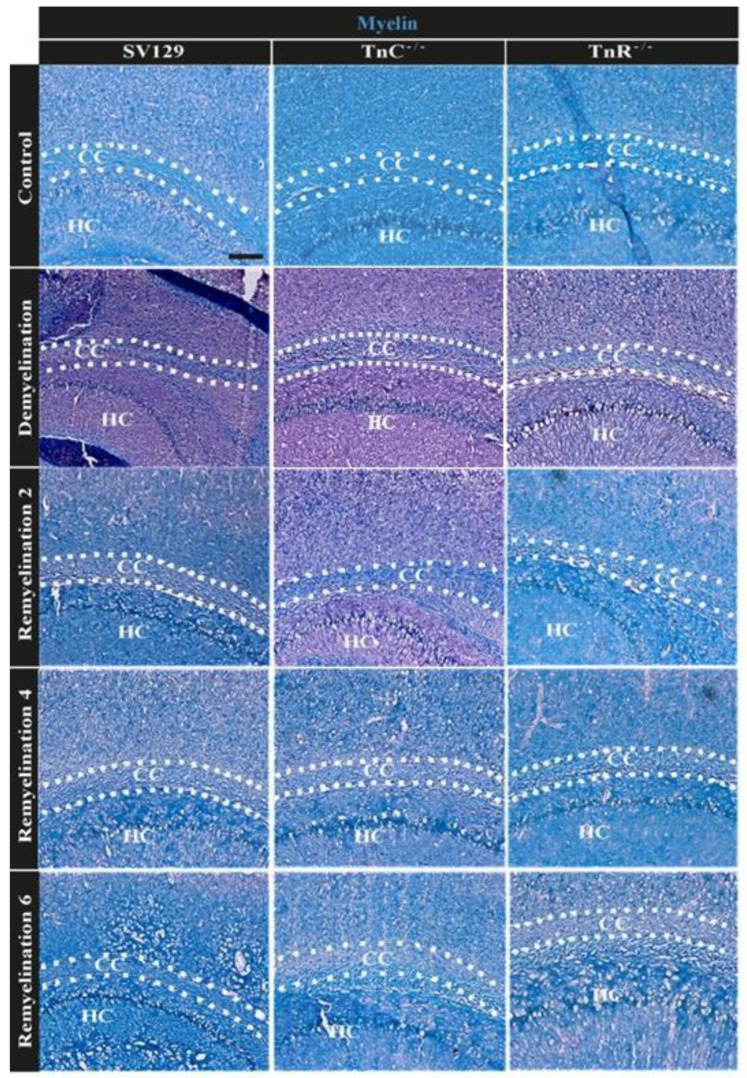
LFB-PAS staining for comparing myelin in the corpus callosum of 129/SV-wildtype, *Tnc^−/−^* and *Tnr^−/−^* mice under control, demyelination, and remyelination conditions. Cryosection of 129/SV wild-type, *Tnc^−/−^* and *Tnr^−/−^* mouse brains were immunohistochemically stained with LFB-PAS in control (C), demyelination (DM), and 2-, 4- and 6 weeks of remyelination (RM2, RM4, and RM6) condition. Myelin was stained blue with LFB, and axons were stained pink via PAS. After 10-weeks of cuprizone administration the blue staining was strongly reduced, indicating efficient removal of myelin. Blue labeled recovered progressively with myelin regeneration, axons were labeled in pink. Cuprizone-induced demyelination as well as remyelination after withdrawal was obvious for the three genotypes. Scale bar 200 µm (N = 1, n ≤ 2).

**Figure 3 cells-11-01773-f003:**
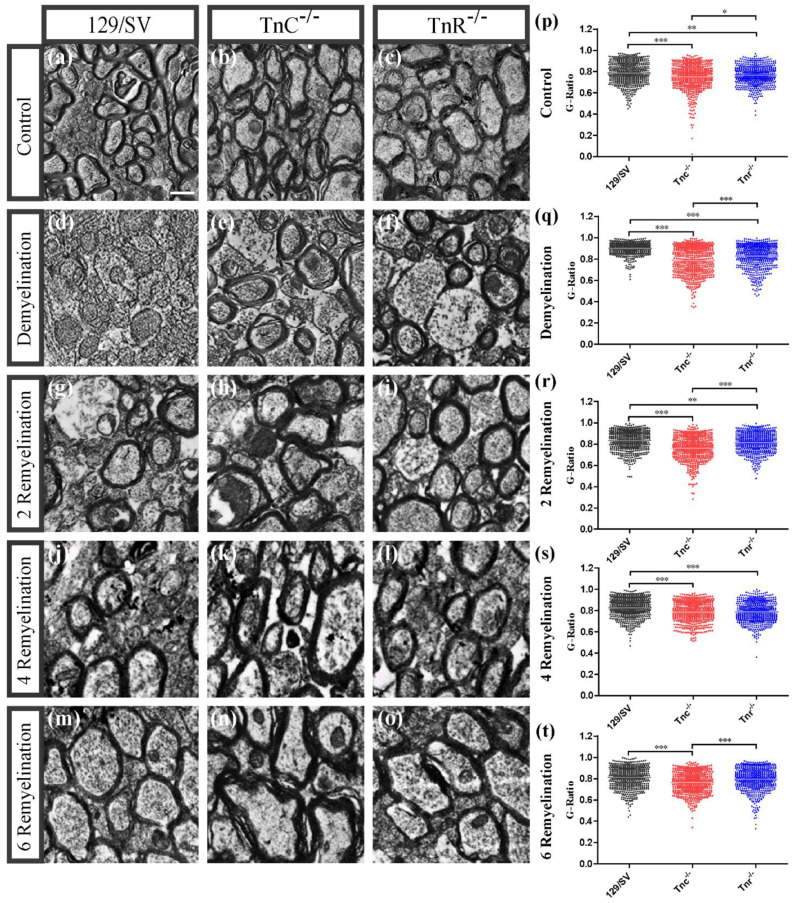
Electron microscopy reveals increased remyelination in *Tnc^−/−^* and *Tnr^−/−^* mice. Eight-to-ten-week-old male mice received either a normal diet as untreated control or a 0.2% cuprizone diet to induce demyelination (**a**–**t**). After 10 weeks mice were either perfused or received a normal diet for 2, 4, or 6 further weeks to allow for remyelination after withdrawal. Cryosections of the corpora callosa (CC) above the hippocampus as an orientation point (HC) from 129/SV, *Tnc^−/−^* and *Tnr^−/−^* brains were collected, processed for electron microscopy and the myelin thickness was determined (**a**–**t**). Scale bar: 1 µm. The results confirmed the cuprizone-induced demyelination (**d**–**f**), although the demyelination in *Tnc^−/−^* and *Tnr^−/−^* mice appeared significantly weaker (**p**). In the untreated control condition (**a**–**c**,**p**) thicker myelin membranes were observable in *Tnc^−/−^* and *Tnr^−/−^* mice as indicated by g-ratios. Furthermore, *Tnc^−/−^* and *Tnr^−/−^* mice displayed more efficient remyelination at each timepoint (**g**–**o**,**r**–**t**). For statistical analysis, the unpaired two-tailed student’s test (*p* ≤ 0.05 *, *p* ≤ 0.01 **, *p* ≤ 0.001 ***) was used. Axon diameters seemed smaller in *Tnc^−/−^* and *Tnr^−/−^* mice. Statistical comparison of the individual groups was carried out with the ANOVA test and Tuckey’s multiple comparison test. (N = 3, n ≤ 594).

**Figure 4 cells-11-01773-f004:**
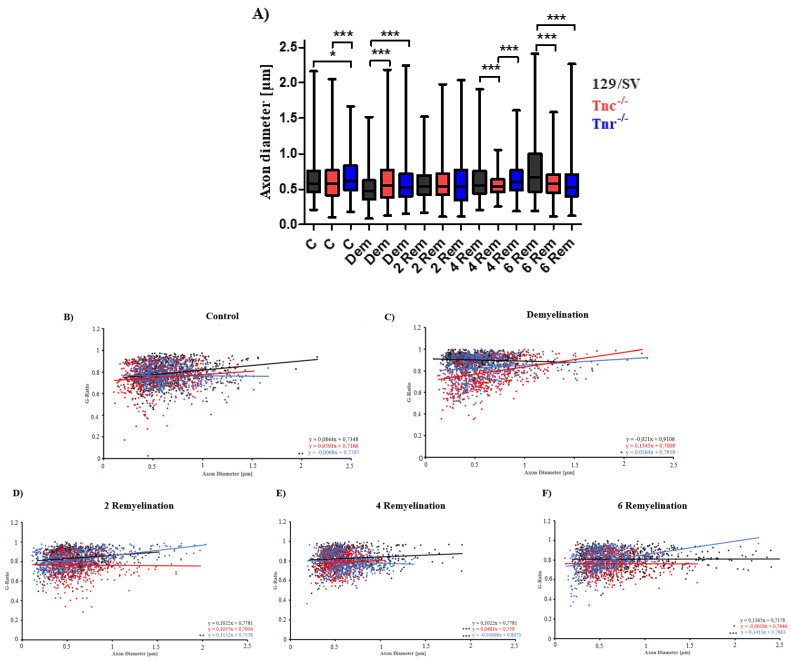
Axon diameter variability of the three genotypes does not seem to have a big influence on the myelination processes. Axon diameter of 129/SV, *Tnc^−/−^* and *Tnr^−/−^* electron microscopy sections of the five different conditions control (C), demyelination (DM), and 2-, 4- and 6 weeks of remyelination (RM2, RM4, and RM6) were analyzed (**A**). In the untreated control condition only in *Tnr^−/−^* mice the axon diameter of the cells was significantly larger than in wildtype mice. Interestingly, during demyelination the determined axon diameters of both *Tnc^−/−^* and *Tnr^−/−^* mice were significantly lower than in wildtype mice. (**B**–**F**) The best fit lines were also obtained by linear regression and differed significantly between *Tnr^−/−^* and wildtype mice in each treatment condition. Statistical analysis was carried out by using the ANOVA and Tukey’s multiple comparison test (*p* ≤ 0.05 *, *p* ≤ 0.01 **, *p* ≤ 0.001 ***). (N = 3, n ≤ 594).

**Figure 5 cells-11-01773-f005:**
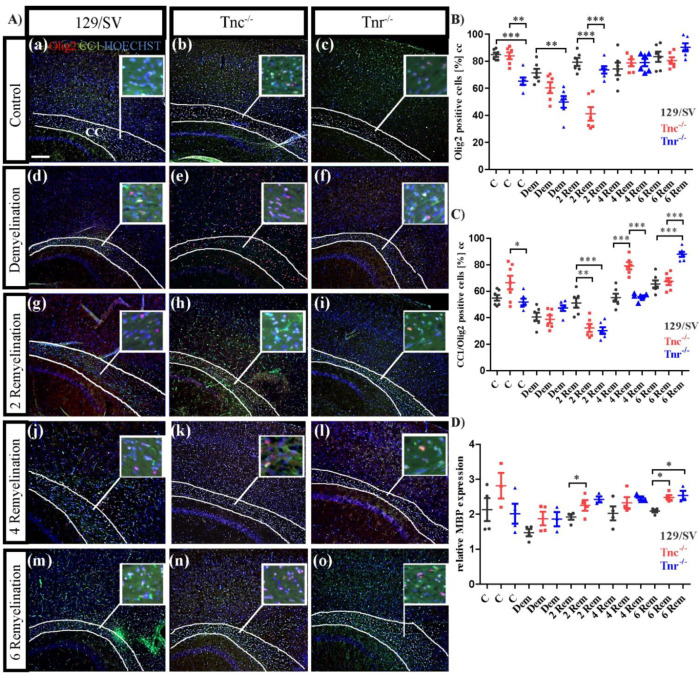
Differential maturation of oligodendrocytes in *Tnc^−/−^* and *Tnr^−/−^* genotypes after cuprizone treatment. (**A**(**a**–**o**)) Sagittal sections of 129/SV, *Tnc^−/−^* and *Tnr^−/−^* mice treated with cuprizone were immunohistochemically stained with antibodies against CC1 (green) and Olig2 (red) in the five different conditions, control (C), demyelination (DM), 2-, 4- and 6 weeks of remyelination (RM2, RM4, and RM6). Hoechst was used as a marker for cell nuclei (blue). Images show the caudal part of the corpus callosum (CC) above the hippocampus (HC), circle triangle squares show a better visualization of the cells. Fewer oligodendrocytes were detected in Tnr*^−/−^* condition in comparison to the 129/SV wildtype and the *Tnc^−/−^* genotype (**A**(**a**–**c**),**B**). CC1 staining was reduced by cuprizone treatment and recovered after 2, 4 and 6 weeks of remyelination (**A**(**d**–**o**)). In the early stage of demyelination after 2 weeks, lowest number of oligodendrocytes was detected in *Tnc^−/−^* mice, and the highest number of immature oligodendrocytes was detected in 129/SV wildtype mice (**A**(**g**–**i**),**C**). RT-PCR analysis of MBP expression revealed a higher myelin expression in both knockouts at the earliest (2 weeks) and latest (6 weeks) stages of recovery, reflecting a more extensive remyelination in *Tnc^−/−^* and *Tnr^−/−^* mice (**D**). Data are presented as mean ± SEM and statistical significance (*p* ≤ 0.05 *, *p* ≤ 0.01 **, *p* ≤ 0.001 ***) was assessed using the ANOVA and Tukey’s multiple comparison test (control, demyelinated, remyelinated). Four animals were used for each group and genotype (N = 4).

**Figure 6 cells-11-01773-f006:**
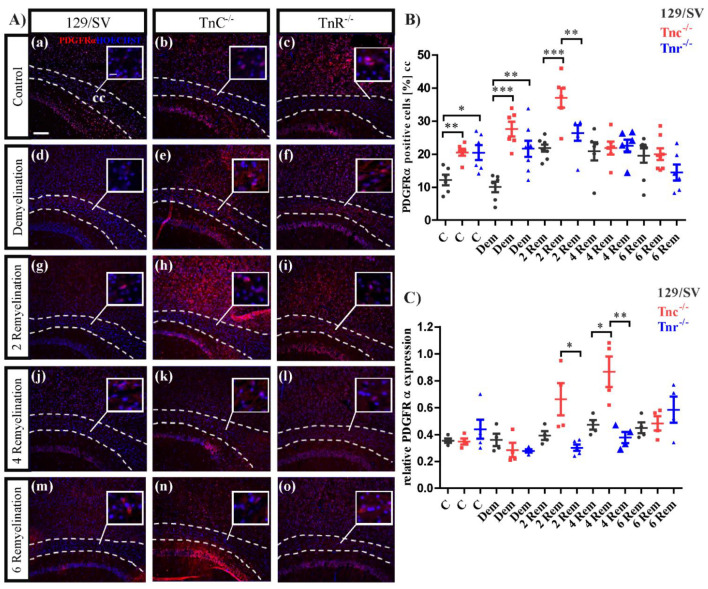
Differential recruitment of oligodendrocyte precursor cells (OPCs) in the *Tnr^−/−^* and *Tnc^−/−^* genotypes. (**A**(**a**–**o**)) Sagittal sections of 129/SV, *Tnc^−/−^* and *Tnr^−/−^* mice exposed to cuprizone were immunohistochemically stained with an antibody against PDGFRα (red) in the conditions, control (C), demyelination (DM), 2-, 4- and 6 weeks of remyelination (RM2, RM4 and RM6). Hoechst was used as a marker for cell nuclei (blue). Images show the caudal part of the corpus callosum (CC) above the hippocampus (HC), circle triangle squares show a better visualization of the cells. Both, *Tnc* and *Tnr* knockout leads to a significant increase of OPCs in the control condition (**A**(**a**–**c**),**B**) and during demyelination (**A**(**d**–**f**),**B**). This effect was also observable in the early stage of recovery, after 2 weeks of remyelination (**A**(**g**–**h**),**B**). However, with ongoing duration of remyelination, no differences between the several genotypes were observable and determinable. Additionally, RT-PCR analysis for the determination of the PDGFRα expression in the three different genotypes in the five different conditions was performed (**C**). The results revealed a significantly higher PDGFRα expression during remyelination stages in *Tnc^−/−^* mice, indicating that Tnc impairs the OPC maturation in general. Data are presented as mean ± SEM and statistical significance (*p* ≤ 0.05 *, *p* ≤ 0.01 **, *p* ≤ 0.001 ***) was assessed using the ANOVA and Tukey’s multiple comparison test (control, demyelinated, remyelinated). Four independent experiments were performed (N = 4).

**Figure 7 cells-11-01773-f007:**
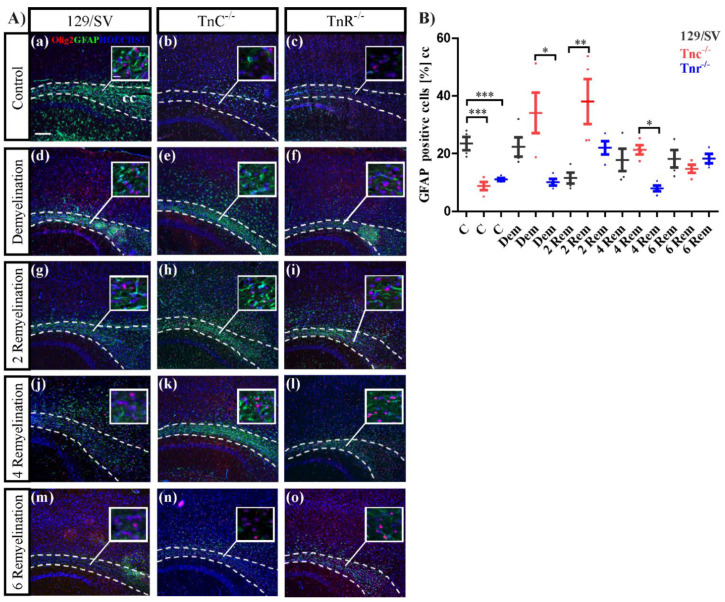
Reactivity of astrocytes in the lesion territory. (**A**(**a**–**o**)) Sagittal sections of the three different genotypes 129/SV, *Tnc^−/−,^* and *Tnr^−/−^* mice in the cuprizone model were immunocytochemically stained with antibodies against GFAP (green) as a marker for astrocytes and Olig2 (red) as a marker for oligodendrocytes. Hoechst dye served as a marker for cell nuclei (blue). Five different conditions were analyzed: control (C), demyelination (DM), 2-, 4- and 6 weeks of remyelination (RM2, RM4 and RM6). Images show the caudal part of the corpus callosum (CC) above the hippocampus (HC), circle triangle squares show a better visualization of the cells. In untreated control conditions, the number of astrocytes was significantly decreased in both knockout (*Tnc^−/−^* and *Tnr^−/−^*) mice (**A**(**a**–**c**),**B**). During demyelination and the early stage of recovery, after 2 weeks of withdrawal of cuprizone treatment, in *Tnc^−/−^* significantly more astrocytes were detectable in comparison to the wildtype mice and untreated control condition (**A**(**a**–**i**),**B**). Otherwise, no significant differences between the three genotypes were visible (**A**(**j**–**o**),**B**). Data are presented as mean ± SEM and statistical significance (*p* ≤ 0.05 *, *p* ≤ 0.01 **, *p* ≤ 0.001 ***) was assessed using the ANOVA and Tukey’s multiple comparison test (control, demyelinated, remyelinated). Four independent experiments were performed (N = 4).

**Figure 8 cells-11-01773-f008:**
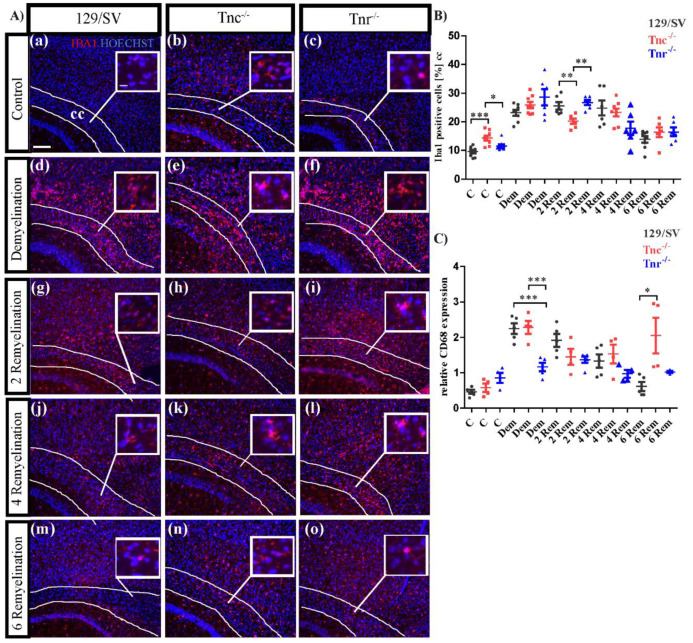
Activation of microglia upon cuprizone treatment. (**A**(**a**–**o**)) Sagittal sections of cuprizone-treated mice were immunohistochemically stained with an antibody against Iba1 to detect microglia (red color). Hoechst was used as a marker for cell nuclei. Five different conditions were analyzed: control (C), demyelination (DM), 2-, 4- and 6 weeks of remyelination (RM2, RM4 and RM6). Images show the caudal part of the corpus callosum (CC) above the hippocampus (HC), circle triangle squares show a better visualization of the cells. Significantly more Iba1 positive cells were detected in *Tnc^−/−^* and *Tnr^−/−^* mice (**A**(**a**–**c**),**B**). During demyelination, the number of Iba1 positive cells was strongly increased in comparison to the control and in the absence of Tnr the number of microglia was significantly increased (**A**(**d**–**f**),**B**). After 2 weeks of remyelination the number of Iba1 positive cells was decreased in *Tnc^−/−^* mice (**A**(**g**–**i**),**B**). This effect contrasts with the results of the latest stage of remyelination after 6 weeks where the number of Iba1 positive cells was significantly increased in *Tnc^−/−^* mice (**A**(**m**–**o**),**B**). RT-PCR analysis of CD68 to monitor the expression pattern of activated microglia was carried out. In the absence of Tnr, more activated microglia were present in the control condition (**A**(**a**–**c**),**C**). In contrast, the CD68 expression during demyelination seemed reduced in *Tnr^−/−^* (**A**(**d**–**f**),**C**). In the absence of Tnr the expression of CD68 was limited and even during demyelination and the first two remyelination stages (2 and 4 weeks) the expression increased only minimally (**A**(**g**–**o**),**C**). After 6 weeks of remyelination, the highest CD68 expression was measured in *Tnc^−/−^*. Data are presented as mean ± SEM and statistical significance (*p* ≤ 0.05 *, *p* ≤ 0.01 **, *p* ≤ 0.001 ***) was assessed using the ANOVA and Tukey’s multiple comparison test for each group (control, demyelinated, remyelinated). Four independent experiments were performed (N = 4).

## Data Availability

The raw data supporting the conclusions of this article will be made available by the authors, without undue reservation.

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
