# Peer review of "The Extracellular Matrix Proteins Tenascin-C and Tenascin-R Retard Oligodendrocyte Precursor Maturation and Myelin Regeneration in a Cuprizone-Induced Long-Term Demyelination Animal Model"

_cells, 2022, doi:10.3390/cells11111773_

Round 1
Reviewer 1 Report
In this manuscript, the authors investigated the hypothesis that extracellular matrix proteins tenascin C and tenascin R have modulatory effects on remyelination. They showed that in an in vitro myelination assay, loss of these proteins resulted in increased myelination. The Cuprizone model of demyelination was used to show that knock-out of these proteins resulted in increased remyelination on the ultrastructural level and elevated oligodendrocyte precursor cell numbers during remyelination in vivo.
The hypothesis raised by the authors is relevant in the field and was tested by appropriate methods in a thorough way. Some aspects of method description and presentation of the results should be improved. Some language issues should be corrected, however, they do not impede comprehension of the text.
Major comments
1) line 381: “Taken together the analysis revealed that in each condition the g-ratios of both knockouts were lower compared to the wildtype condition. This suggested that the ablation of Tnc as well as Tnr favored remyelination”
After demyelination, different levels of g-ratios where observed in the three mouse strains, probably resulting from slightly different vulnerabilities towards Cuprizone intoxication. Therefore, differences in g-ratios after remyelination might be due to different levels of myelination at the beginning of the remyelination period rather than a different remyelination capacity. For this reason, the authors’ conclusion of elevated remyelination capacities in both knock-out strains is not completely convincing.
2) For histological analyses in the Cuprizone model the choice of cutting level and region of interest is of high importance for comparability and reproducibility because of the regional heterogeneity of Cuprizone-induced demyelination. At the level of the hippocampus, demyelination is more severe close to the midline of the corpus callosum than in its lateral portion. Please include details on the chosen region and the level of sagittal cutting. Has the analysis been performed in regions of severe or mild demyelination?
3) Fig 5/6: The image quality is moderate. Especially the red signal in figure 5 is hard to detect and the more informative close-ups are rather small. The figures would benefit from larger high magnification views instead of low magnification overviews of the corpus callosum.
4) Analysis of remyelination in the Cuprizone model would be more convincing by providing an analysis of the myelination status by immunohistochemical staining of myelin markers (e.g. PLP, MBP, …) in addition to the ultrastructural analysis.
Minor comments
line 53: “With regard to the clinical picture of MS we focused on toxicity-based demyelination, namely the cuprizone model.”
Most clinical features of MS are not covered in the Cuprizone model, especially regarding autoimmune inflammatory aspects. Please specify why the Cuprizone model was chosen.
line 267: Please specify “In individual cases, …”
line 270: RPTP-β was not investigated in this study.
line 275: “All RT-PCR results were analyzed with the rectangle tool ImageJ. In this context, the mean gray value of each sample was measured, the background was subtracted, and the resulting values were set in relation to the actin signal.”
Please specify what gray values have been measured. I would have expected Ct values as results of qRT-PCR.
line 300: Were all groups compared to every other group or was each group compared to the respective wildtype group? Please specify.
line 313: “Both the average number of fibers myelinated by a single cell which identifies the amount of myelin formed, as well as the numbers of sheaths extended per single oligodendrocyte were analyzed (Fig. 1C-F).”
Please clarify the difference between number of sheaths and number of fibers per single oligodendrocyte. The methods section states that the cumulative sheath length per oligodendrocyte was measured.
line 328/Fig 1: A wildtype control with BDNF treatment is missing. An explanation would be helpful to evaluate the experimental set-up. Statistical comparison of the untreated knock-out group with the BDNF-treated knock-out group seems more appropriate than comparison of the BDNF-treated knock-out with untreated control. Indicators of significance are missing in Fig 1D. The identity of the cells in Fig 1B in the upper and lower right picture is unclear, are they wildtype or Tnc-/- or Tnr-/-?
line 355: “To test this aspect, both wildtype and knockout mice were subjected to cuprizone-mediated demyelination for a period of 10 weeks, modelling chronic demyelination [14,15].”
What are the reasons to choose a model of chronic demyelination over a model of acute demyelination? For remyelination studies, shorter Cuprizone intoxication periods are often preferred because remyelination capacities are larger after acute demyelination.
line 369: Figure 4 is referred to before figures 2 and 3. Changing the order of figures seems appropriate.
line 413/Fig 3: For better understanding of the analysis of axon diameter and g-ratio, it would be beneficial to provide the reader with some explanation on how to interpret the similarity or differences of the best fit lines. Additionally, identical scaling of the diagrams would facilitate comparisons between conditions.
line 538: “Therefore, we investigated microglia using the markers Iba1 and CD68 expression in our demyelination model”
Discrimination of microglia and invading peripheral monocytes/macrophages is not possible with these markers. A remark on this limitation would be appropriate.
Fig 5-8: The diagrams tend to be confusing due to different shapes of datapoints. Uniform shapes would be helpful.
Author Response
Reviewer 1
Summary:
In this manuscript, the authors investigated the hypothesis that extracellular matrix proteins tenascin C and tenascin R have modulatory effects on remyelination. They showed that in an in vitro myelination assay, loss of these proteins resulted in increased myelination. The Cuprizone model of demyelination was used to show that knock-out of these proteins resulted in increased remyelination on the ultrastructural level and elevated oligodendrocyte precursor cell numbers during remyelination in vivo.
Answer: We thank the Reviewer for reviewing our manuscript and summarizing the highlights of our results. We hope that the revision improves the quality of our manuscript according to the reviewer´s comments.
Major comments
- “After demyelination, different levels of g-ratios where observed in the three mouse strains, probably resulting from slightly different vulnerabilities towards Cuprizone intoxication. Therefore, differences in g-ratios after remyelination might be due to different levels of myelination at the beginning of the remyelination period rather than a different remyelination capacity. For this reason, the authors’ conclusion of elevated remyelination capacities in both knock-out strains is not completely convincing.”
Answer: We thank the reviewer for this comment. We can definitely conclude that the differences in the g-ratios are due to the different genotypes during demyelination. It is known that the cuprizone treatment induces a demyelination in the corpus callosum already after 5 to 6 weeks [1]. For our model we have increased the duration of cuprizone treatment to ensure that the otherwise beneficial effects of the knockout compromise a successful demyelination. When we compare the g-ratios of the different genotypes it become clear that the cuprizone model is also working in the knockout mice. However, we include a figure to the supplement part that documents the changes of bodyweight under the different conditions. It appears that the Tnc-/- mice experience weight loss as demyelination increases, yet less than other genotypes.
Answer: We thank the reviewer for pointing out the differences of myelin sheaths upon demyelination between the different mouse lines. Although the lines have comparable SV129 backgrounds the independent breeding of the colonies may have resulted in small differences. We carefully examined the literature and found that both the Tnc-/- and the Tnr-/- knockout lines apparently do not display visible myelination deficits. This does not exclude a differential sensitivity to Cuprizone treatment. Known effects concern the roles of Tnc in regulating proliferation and migration, and these aspects are themes in our discussion.
“The analysis of g-ratios that present the relation of myelin sheath thickness to axon diameter indicated differences between the mouse lines under study, both under control conditions (Fig. 2B) and after demyelination (Fig. 2C). Although the lines have comparable SV129 backgrounds the independent breeding of the colonies may have resulted in small differences. Previous studies carefully worked out that both the Tnc-/- and the Tnr-/- knockout lines apparently do not display visible myelination deficits [2,3].”
“This does not exclude a differential sensitivity to Cuprizone treatment. Interestingly, known effects concern the roles of Tnc in regulating proliferation and migration of OPCs [4]. Furthermore, an anti-apoptotic effect of Tnc for oligodendrocytes in vitro has been reported [5]. Less is known in this regard about Tnr. Therefore, the question remains open whether some of the differences of de- and remyelination observed are due to a differential susceptibility of the mouse lines towards Cuprizone treatment.”
“..(Suppl. Fig. 2), once a week cuprizone treated mice also received a normal diet to minimize the severity of intoxication.”
“All in all, 12 mice per genotype were used for each condition, so that in total 180 mice were included in this study. However, not all individual animals were analyzed.”
Supplementary Material and Methods:
“Supplemental Figure S2: Weight documentation during, without and after cuprizone treatment in the different genotypes (SV/129, Tnc-/- and Tnr-/-). Weight changes of the mice were documented three times a week. Under control conditions weight increased steadily over time in each genotype. During demyelination the weight of the mice decreased over time and was indicative of successful demyelination as a result of cuprizone treatment. Once a week cuprizone treated mice received a normal diet to minimize the severity of intoxication. Therefore, interim peaks were detectable. With ongoing remyelination time the weight of the mice increased in each genotype, which is an indicator for successful remyelination. “
- For histological analyses in the Cuprizone model the choice of cutting level and region of interest is of high importance for comparability and reproducibility because of the regional heterogeneity of Cuprizone-induced demyelination. At the level of the hippocampus, demyelination is more severe close to the midline of the corpus callosum than in its lateral portion. Please include details on the chosen region and the level of sagittal cutting. Has the analysis been performed in regions of severe or mild demyelination?
Answer: We decided to analyze the regions of the caudal and rostral part of the corpora callosa, however we could not detect any differences between these regions. Therefore, we pooled the data, and analyzed the whole corpus callosum in each condition and genotype in this way. The corpus callosum is known to be the part of the severe demyelination and this is also in conclusion with our observations.
“Afterwards, cryosections of 14 μm were cut of one hemisphere on a cryostat and stored at −20°C. Here, the area on sagittal sections was focused on the corpus callosum (CC) above the hippocampus (HC) (according to Bregma: lateral 0.32 – 0.48mm), where myelination is highest”.
“Here, the caudal part as well as the rostral part of the corpus callosum was analyzed in more detail, because these are the most affected areas. The corpus callosum is known to be subject to severe demyelination [6,7].”
3) Fig 5/6: The image quality is moderate. Especially the red signal in figure 5 is hard to detect and the more informative close-ups are rather small. The figures would benefit from larger high magnification views instead of low magnification overviews of the corpus callosum.
Answer: We thank the reviewer for pointing to this deficit and have revised the figure 5 accordingly.
4) Analysis of remyelination in the Cuprizone model would be more convincing by providing an analysis of the myelination status by immunohistochemical staining of myelin markers (e.g. PLP, MBP, …) in addition to the ultrastructural analysis.
Answer: We thank the reviewer for this comment and agree that these would be reasonable additional experiments. However, in our present study we have analyzed MBP expression by using RT-PCR analysis and found that both tenascins seem to limit MBP expression during remyelination. It is known that MBP is one of the most important myelin proteins and a limiting factor of myelin membrane formation [8,9]. So we think that it represents a sufficient biomarker in the context of our investigation. As the focus was on the role of tenascins we would leave the analysis of mature myelin markers to a follow-up study.
“MBP represents a well-established biomarker for the maturation of myelin membranes.”
Minor comments
Most clinical features of MS are not covered in the Cuprizone model, especially regarding autoimmune inflammatory aspects. Please specify why the Cuprizone model was chosen.
Answer: Thank you for your comment. We have chosen the cuprizone model, because it is an excellently established model to analyze remyelination. We postulated that both tenascins inhibit the remyelination process. EAE is a model which is usually used to analyze autoimmune-mediated demyelination and definitely closer to the pathophysiology of the disease of multiple sclerosis. However, the EAE model involves the activation of lymphocytes and the blood brain barrier is destroyed, which renders interpretation difficult. In our approach the emphasis was on the impact of tenascins on the remyelination process, which can be investigated using the cuprizone model.
“The cuprizone model places the focus on the remyelination process in the absence of activated T-cells. Along this path we intended to analyze a more severe demyelination followed by a longer course of remyelination so that the influence of both tenascins on remyelination efficiency in what appears closer to a chronic setting could also be determined.“
line 267: Please specify “In individual cases, …”
Answer: Thank you for pointing this out. We have rewritten the sentence.
“Moreover, in each condition four animals with the following genes were investigated: platelet-derived growth factor receptor A (PDGFRα), myelin-basic protein (MBP) and ionized calcium-binding adapter molecule 1 (Iba1).”
line 270: RPTP-β was not investigated in this study.
Answer: We thank the reviewer for this hint and deleted the mention of RPTP-b.
line 275: “All RT-PCR results were analyzed with the rectangle tool ImageJ. In this context, the mean gray value of each sample was measured, the background was subtracted, and the resulting values were set in relation to the actin signal.”
Answer: We thank you for this comment and explain the procedure in more accuracy. We carried out RT-PCR analysis in the presents study. We removed the inappropriate description of qPCR.
line 300: Were all groups compared to every other group or was each group compared to the respective wildtype group? Please specify.
Answer: We thank the reviewer for this question. We compared the mutants to the wildtype group in each condition. Furthermore, we have included analysis with ONE way ANOVA and following Tukey´s test in some cases to analyze all three genotypes in the several conditions comparatively to each other. The test used are indicated in the figure legends.
„The significances were determined using ONE-way ANOVA with subsequent Tukey´s multiple comparisons test. In pairwise comparisons also the unpaired two-tailed Student’s T-test was used. In some cases the Kruskal-Wallis test followed by Dunn´s multiple comparison test was applied. The tests used are indicated in the figure legends.“
line 313: “Both the average number of fibers myelinated by a single cell which identifies the amount of myelin formed, as well as the numbers of sheaths extended per single oligodendrocyte were analyzed (Fig. 1C-F).”
Answer: We thank the reviewer for this suggestion and changed the description. In these cases, the average sheath length as well as the number of sheaths one oligodendrocyte built was analyzed.
Line 363/364: “..the average sheath length of fibers”
line 328/Fig 1: A wildtype control with BDNF treatment is missing. An explanation would be helpful to evaluate the experimental set-up. Statistical comparison of the untreated knock-out group with the BDNF-treated knock-out group seems more appropriate than comparison of the BDNF-treated knock-out with untreated control. Indicators of significance are missing in Fig 1D. The identity of the cells in Fig 1B in the upper and lower right picture is unclear, are they wildtype or Tnc-/- or Tnr-/-?
Answer: We thank the reviewer these comments and explain our procedure in more detail, furthermore we improved the figure. We decided to analyzed the influence of BDNF on the knockout cultures in a feasibility study to see whether myelin formation by the mutants can be boosted by the addition of purified BDNF. The cells in Fig. 1B in the upper and lower right are knockout cells. We added specifications to figure 1.
line 355: “To test this aspect, both wildtype and knockout mice were subjected to cuprizone-mediated demyelination for a period of 10 weeks, modelling chronic demyelination [14,15].”
What are the reasons to choose a model of chronic demyelination over a model of acute demyelination? For remyelination studies, shorter Cuprizone intoxication periods are often preferred because remyelination capacities are larger after acute demyelination.
Answer: We thank the author for this question and try to explain it in more detail. We wanted to analyze, whether the longer duration of cuprizone treatment led to bigger difference between the genotypes during demyelination. In our previous study, after 6 weeks of cuprizone treatment only few differences between the genotypes were recognizable. Therefore, we attempted to mimic a more chronic mode of demyelination and to determine the influence of both tenascins on remyelination efficiency in this context.
“The cuprizone model places the focus on the remyelination process in the absence of activated T-cells. Along this path we intended to analyze a more severe demyelination followed by a longer course of remyelination so that the influence of both tenascins on remyelination efficiency in what appears closer to a chronic setting could also be determined.“
line 369: Figure 4 is referred to before figures 2 and 3. Changing the order of figures seems appropriate.
Answer: We thank the author for this hint and adapted the order of figures accordingly.
Line 369-373: “To determine whether the cuprizone induced demyelination as well as the remyelination efficiency after cuprizone withdrawal are working we performed LFB-PAS staining. As expected, the corpora callosa were stained in blue in the untreated control condition and the staining appeared weaker upon demyelination (Fig. 2).”
line 413/Fig 3: For better understanding of the analysis of axon diameter and g-ratio, it would be beneficial to provide the reader with some explanation on how to interpret the similarity or differences of the best fit lines. Additionally, identical scaling of the diagrams would facilitate comparisons between conditions.
Answer: Thank you for pointing to the necessity to adjust the scaling and to explain the meaning of the slopes. We have adjusted the scaling in Fig. 4 B-F, as demanded.
“…g-ratios obtained against multiple axon diameters (Fig. 4 B-F). The linear regression slopes correlate axon diameter with g-ratios.”
“Thereby, an ascending slope indicates that the myelin sheath grows at a slower rate than the axon diameter while a descending slope reflects an increased growth of the myelin sheath with growing axon diameter. Concerning this parameter our results confirmed that linear regression lines differed significantly especially with regard to the Tnr-/- tissue (Fig. 4B-F). Interestingly, after 4 weeks of remyelination the axons of the Tnc-/- and Tnr-/- mouse lines have acquired relatively larger myelin sheaths than the wildtype (Fig. 4E). According to this parameter, the myelin sheaths in relation to the axon diameter were thinner in the wildtype than in the mutants also in the samples after demyelination and two weeks of remyelination (Fig. 4 C,D). “
line 538: “Therefore, we investigated microglia using the markers Iba1 and CD68 expression in our demyelination model”
Discrimination of microglia and invading peripheral monocytes/macrophages is not possible with these markers. A remark on this limitation would be appropriate.
Answer: We thank the author for this suggestion and have revised parts of our description. However, we think that these are good markers for the analysis of the response of effector cells of the central nervous system.
“..we investigated the local immune response including microglia and macrophages”
“The microglia and macrophage compartment were investigated further using the marker CD68 that was evaluated by RT-PCR analysis (Fig. 8C). CD68 is a highly glycosylated glycoprotein which is expressed by mononuclear phagocytes and used to detect the response of activated microglia as well as macrophages that accumulate in acute lesions [10].”
Fig 5-8: The diagrams tend to be confusing due to different shapes of datapoints. Uniform shapes would be helpful.
Answer: We thank the reviewer for this comment and have made an effort to improve the clarity of our figures. In particular, we have unified the presentation of the data points and the color code.
[JB1]Matsushima PMID: 11145196
Mason PMID: 10900072

Reviewer 2 Report
The paper submitted by Bauch and Faissnet is based on the hypothesis that tenascins, mainly tenascin C (Tnc) and tenascin R (Tnr) interfere with remyelination in vivo. The main argument for this assumption is the previous work reported that tenascins interfere with remyelination processes in an ex vivo cerebellar slice cultures. The authors make reference to an unpublished paper, (reference [31]). Now, this paper has been published, and I have carefully studied it. I found that experiments were not only conducted ex vivo with cerebellar explants, but some experiments were designed to include in vivo. Moreover, to address whether Tnc and Tnr are involved in myelin membrane formation, Bauch et al. (Front Cell Dev Biol. 2022 Mar 15;10:819967. doi: 10.3389/fcell.2022.819967) used cuprizone-based demyelination animal models. Their experiments were performed on both control mice (wild type SV129) and Tnc- and Tnr-deficient mice, exactly the same phenotypes of mice as used in this submitted paper. Therefore, taking into account the recently published paper by Bauch et al., the current manuscript requires in-depth modifications throughout the manuscript. Additionally, I do have several major comments that should be addressed to better clarify the results.
In the materials and methods section, some relevant data were omitted. For example, in total how many mice were used in the whole study. And more specifically, how many mice were used in each group to study cuprizone induced demyelination, including adequate control groups.
Line 210 – The authors stated that the weight of all animals was recorded three times a week. The results of these records have not been described. Did administration of cuprizone affect the weight of mice?
Line 224 – The brain segment containing the corpus callosum was cut. Sagittal sections of 14 µm thickness were collected and stained. The procedure of brain sectioning and histological or immunohistochemical staining should be described in detail, including the approximate thickness of the brain segment that was cut as reported in the subsection with subheading 2.10 Electron microscopy. The number of collected series of brain sections should also be given.
Line 301 - In the subsection 2.11 Statistics details of statistical tests have been provided, including One Way ANOVA followed by Tukey’s multiple comparison test. However, the results of these tests have not been reported in the results section.
Line 328 – To test whether downregulation of tenascin genes affects the myelination process oligodendrocytes cultured from the homozygotes and heterozygotes mice were tested in a fibre myelination assay. Next, the neurotrophin BDNF was added to cell cultures which resulted in a significant increase in the number of sheaths formed per single cell. For each tenascin, 3 different groups were studied, and to show differences between groups ANOVA should be appropriate statistical test here. Simple description of number of sheaths per individual cells is not sufficient. Other values such as the degrees of freedom and the confidence level of the Type I error should be provided. Similar statistical tests should be made in line with these principles, that is to say in all cases when more than 2 groups are analysed.
The results described in the subsection 3.3 Tnc and Tnr modulate recruitement of OPCs to and their maturation in myelin lesions were already published in a previous paper by Bauch et al. [31], except data from the experimental group “6 weeks of remyelination RM6”. However, I don’t think that these results are of any importance here and that they should be included in this manuscript. Another concern is the lack of description of double labeling (Olig2 and CC1) immunohistochemistry and analysis of these data. How many sections were immunotained in each group, how many double labeled cells were analysed and quantified?
As a complement, an additional marker PDGFRα of oligodendrocyte precursor cells (OPC) was used. What was it driven by? The use of this marker would make sense if double labeling for Olig2 and PDGFRα immunohistochemistry was performed. This allows to study whether there are differences in the distribution of OPC cells in the corpus callosum.
The results reported in the subsections 3.4. The loss of Tnc enhances astrocyte reactivity in cuprizone-induced CNS lesions and 3.5. Tenascins modulate microglia and leucocytes in cuprizone-induced lesions should be analysed using ANOVA followed by post-hocs tests.
Images presented in Figures 5 to 8 are not of good quality and should be improved.
Author Response
Reviewer 2
The paper submitted by Bauch and Faissner is based on the hypothesis that tenascins, mainly tenascin C (Tnc) and tenascin R (Tnr) interfere with remyelination in vivo. The main argument for this assumption is the previous work reported that tenascins interfere with remyelination processes in an ex vivo cerebellar slice cultures.
Answer: We thank the author summarizing our data and for the interest in our recently published study.
In the materials and methods section, some relevant data were omitted. For example, in total how many mice were used in the whole study. And more specifically, how many mice were used in each group to study cuprizone induced demyelination, including adequate control groups.
Answer: Thank you for making us aware this point. For each condition 12 animals were used, in the end 60 animals for each genotype were analyzed (12x untreated control, 48x cuprizone diet from which 36 received a normal diet to allow for remyelination after cuprizone induced demyelination). We have specified these aspects in the material and methods section.
“All in all, 12 mice per genotype were used for each condition, so that in total 180 mice were included in this study. However, not all individual animals were analyzed.”
Line 210 – The authors stated that the weight of all animals was recorded three times a week. The results of these records have not been described. Did administration of cuprizone affect the weight of mice?
Answer: We thank the reviewer for this suggestion and added a figure to the supplement part in which the weight changes of all animals during the experiment is shown.
“Supplemental Figure S2: Weight documentation during, without and after cuprizone treatment in the different genotypes (SV/129, Tnc-/- and Tnr-/-). Weight changes of the mice were documented three times a week. Under control conditions weight increased steadily over time in each genotype. During demyelination the weight of the mice decreased over time and was indicative of successful demyelination as a result of cuprizone treatment. Once a week cuprizone treated mice received a normal diet to minimize the severity of intoxication. Therefore, interim peaks were detectable. With ongoing remyelination time the weight of the mice increased in each genotype, which is an indicator for successful remyelination. “
Line 224 – The brain segment containing the corpus callosum was cut. Sagittal sections of 14 µm thickness were collected and stained. The procedure of brain sectioning and histological or immunohistochemical staining should be described in detail, including the approximate thickness of the brain segment that was cut as reported in the subsection with subheading 2.10 Electron microscopy. The number of collected series of brain sections should also be given.
Answer: We thank the author for this suggestion and tried to explain the procedure how we performed the analysis of the cryosections in more detail.
“ …according to Bregma: lateral 0.32 – 0.48mm)”.
“Here, the caudal part as well as the rostral part of the corpus callosum was analyzed in more detail, because these were the most affected areas. The corpus callosum is known to be subject to severe demyelination [6,7].”
“For each condition at least 8 sections were collected.”
Line 301 - In the subsection 2.11 Statistics details of statistical tests have been provided, including One Way ANOVA followed by Tukey’s multiple comparison test. However, the results of these tests have not been reported in the results section.
Answer: We thank the reviewer for making us aware to this point and have changed it, by using ONE way ANOVA and following Tukey´s test to analyze all three genotypes in the several conditions in comparison to each other.
„The significances were determined using ONE-way ANOVA with subsequent Tukey´s multiple comparisons test. In pairwise comparisons also the unpaired two-tailed Student’s T-test was used. In some cases the Kruskal-Wallis test followed by Dunn´s multiple comparison test was applied. The tests used are indicated in the figure legends.“
Line 328 – To test whether downregulation of tenascin genes affects the myelination process oligodendrocytes cultured from the homozygotes and heterozygotes mice were tested in a fibre myelination assay. Next, the neurotrophin BDNF was added to cell cultures which resulted in a significant increase in the number of sheaths formed per single cell. For each tenascin, 3 different groups were studied, and to show differences between groups ANOVA should be appropriate statistical test here. Simple description of number of sheaths per individual cells is not sufficient. Other values such as the degrees of freedom and the confidence level of the Type I error should be provided. Similar statistical tests should be made in line with these principles, that is to say in all cases when more than 2 groups are analysed.
Answer: We thank the reviewer for this question and suggestion. In this experiment we only compared the knockout condition to the wildtype condition and the knockout with purified BDNF to the knockout condition because we intended to analyze the influence of the addition purified BDNF to the knockout. Several publications are available concerning the influence of BDNF on oligodendrocytes in general.
The results described in the subsection 3.3 Tnc and Tnr modulate recruitement of OPCs to and their maturation in myelin lesions were already published in a previous paper by Bauch et al. [31], except data from the experimental group “6 weeks of remyelination RM6”. However, I don’t think that these results are of any importance here and that they should be included in this manuscript. Another concern is the lack of description of double labeling (Olig2 and CC1) immunohistochemistry and analysis of these data. How many sections were immunotained in each group, how many double labeled cells were analysed and quantified?
Answer: We thank the author for this comment. In our opinion the data from Bauch et al provide a fundament for this study in that it showed mainly effects of tenascins in an ex vivo approach. The in vivo studies were much less detailed. The novel aspects in our present work are that we challenged the CNS for a longer period with cuprizone, mimicking a chronic demyelination. And second we analyzed the in vivo situation by electron microscopy. No study so far has analyzed the impact of tenascin genes in the cuprizone model on the ultrastructural level. Thereby, we offer new insights in myelin plasticity.
“For each condition and animal at least 2 sections were analyzed and and at least 500 cells were examined.“
As a complement, an additional marker PDGFRα of oligodendrocyte precursor cells (OPC) was used. What was it driven by? The use of this marker would make sense if double labeling for Olig2 and PDGFRα immunohistochemistry was performed. This allows to study whether there are differences in the distribution of OPC cells in the corpus callosum.
Answer: PDGFRα is an established marker of OPCs that we used to determine the number of precursor cells. We performed Olig2/CC1 double immunolabeling to follow the maturing oligodendrocytes. So in this strategy we obtained the mature oligodendrocytes as well as their precursors.
“…which is an established marker for OPCs.”
The results reported in the subsections 3.4. The loss of Tnc enhances astrocyte reactivity in cuprizone-induced CNS lesions and 3.5. Tenascins modulate microglia and leucocytes in cuprizone-induced lesions should be analysed using ANOVA followed by post-hocs tests.
Answer: We thank the reviewer for this comment. However, we decided to use the unpaired two-tailed student´s t-test for each group, because this test takes as input 2 sample sets that are independent of each other, and the test’s outputs follow a T-distribution.
Images presented in Figures 5 to 8 are not of good quality and should be improved.
Answer: We revised our figures and have made an effort to improve the quality.

Round 2
Reviewer 1 Report
Thank you for revising the manuscript and clarifying the raised concerns.
Author Response
We thank the reviewer for helpful comments and the favourable receipt of our revision,
Reviewer 2 Report
The authors addressed most of comments to my satisfaction. However, they did not respond to the following comment: “Taking into account the recently published paper by Bauch et al., the current manuscript requires in-depth modifications throughout the manuscript”. Namely, I expected a more detailed description of the data from Bauch et al. (2022) in the introduction section. Interestingly, the authors stated that “Bauch et al. provide a fundament for this study in that it showed mainly effects of tenascins in an ex vivo approach. The in vivo studies were much less detailed“. Nonetheless, Bauch et al. performed in vivo experiments, and they “established cuprizone-based acute demyelination to analyze the remyelination behavior after cuprizone withdrawal in SV129, Tnc−/−, and Tnr−/− mice”. Therefore, the role of tenascin proteins (Tnc and Tnr) in cuprizone-mediated acute and chronic demyelination should be discussed in the discussion section. The authors should also explain why chronic demyelination was induced by cuprizone administration for 10 weeks. For this purpose, researchers perform a 12 week or longer period of cuprizone intoxication (for review, see Zhan et al., The Cuprizone Model: Dos and Do Nots. Cells. 2020 Apr; 9(4): 843.).
The second issue relates to statistics. To study remyelination after withdrawal of cuprizone, 3 different periods, 2, 4 and 6 weeks were examined for each genotype of mice. As the authors wrote “Lines 402-403: With increasing duration of remyelination, the myelin membranes progressively recovered”. To analyze whether there is a significant difference between the 3 groups, ANOVA should be an appropriate statistical test.
Author Response
Reviewer 2:
The authors addressed most of comments to my satisfaction. However, they did not respond to the following comment: “Taking into account the recently published paper by Bauch et al., the current manuscript requires in-depth modifications throughout the manuscript”. Namely, I expected a more detailed description of the data from Bauch et al. (2022) in the introduction section. Interestingly, the authors stated that “Bauch et al. provide a fundament for this study in that it showed mainly effects of tenascins in an ex vivo approach. The in vivo studies were much less detailed“. Nonetheless, Bauch et al. performed in vivo experiments, and they “established cuprizone-based acute demyelination to analyze the remyelination behavior after cuprizone withdrawal in SV129, Tnc−/−, and Tnr−/− mice”. Therefore, the role of tenascin proteins (Tnc and Tnr) in cuprizone-mediated acute and chronic demyelination should be discussed in the discussion section. The authors should also explain why chronic demyelination was induced by cuprizone administration for 10 weeks. For this purpose, researchers perform a 12 week or longer period of cuprizone intoxication (for review, see Zhan et al., The Cuprizone Model: Dos and Do Nots. Cells. 2020 Apr; 9(4): 843.).
Answer: We thank the reviewer for this comment and thoughtful suggestions and try to explain our reasons in more detail.
Typically, mice are treated for 10 until 12 weeks to induce demyelination in a chronic way [1-3]. However, it could also be shown that after a 12-week cuprizone treatment remyelination is very sparse, resulting in a model of chronic demyelination [4] and remyelination that in some cases is insufficient, or even fails [5-7]. To ensure the successful remyelination, we therefore decided to treat the mice for 10 weeks with cuprizone to mimic a more chronic course of demyelination.
Currently, a chronic demyelination is induced by a 12-week cuprizone treatment [2,3]. However, in these cases the remyelination fails or is insufficient [5-8] due to the reduced expression of Trem2 resulting in an impaired clearance of myelin debris from microglia. Therefore, we opted for a 10-week cuprizone treatment to mimic a chronic course of demyelination.
The second issue relates to statistics. To study remyelination after withdrawal of cuprizone, 3 different periods, 2, 4 and 6 weeks were examined for each genotype of mice. As the authors wrote “Lines 402-403: With increasing duration of remyelination, the myelin membranes progressively recovered”. To analyze whether there is a significant difference between the 3 groups, ANOVA should be an appropriate statistical test.
Answer: We thank the reviewer for the remark and comments. In a first step we confirmed the successful de- and remyelination by histochemical stain, as shown in figure 2.
As expected, the corpora callosa were stained in blue in the untreated control condition and the staining appeared weaker upon demyelination (Fig. 2).
Answer: A quantitative analysis was performed on the basis of extensive measurements based on electron microscopy. We now include an in-depth analysis and statistical comparison of the g-ratios that are displayed in detail in the supplemental figure 1.
Answer: With regard to the statistical evaluation we performed a first analysis with the Kruskall-Wallis test which is a non-parametric test that can be applied to more than two independent groups. In response to the reviewer we recalculated the significance levels using ANOVA followed by Tukey’s multiple comparison test.
Statistical comparison of the individual groups was carried out with the ANOVA test and Tuckey’s multiple comparison test.
By comparing the g-ratios of the several remyelination conditions to demyelination, it became clear that remyelination is successful as the g-ratios in all three remyelination conditions (RM2, RM4 and RM6) were significantly reduced in wildtype mice (Suppl. Fig. 1A) and decreased with ongoing remyelination time (g-ratio 129/SV 2R vs 6R: p<0.0001; 4R vs 6R: p=0.0023). This effect could also be observed in the knockout mice (g-ratios Tnc-/- DM vs 2R: p=0.004; DM vs 6R: p=0.0034; 2R vs 4R: 0.0028; 4R vs 6R: p=0.0024 and for Tnr-/- 2R vs 4R: p<0.0001; 2R vs 6R: p=0.0023) (Suppl. Fig. 1B, 1C). However, in Tnr-/- mice the effective remyelination only evolves at the 4th week with a significant difference (g-ratios Tnr-/- D vs 4R: p<0.0001; D vs 6R: p<0.0001) (Suppl. Fig. 1C).
References
- Blakemore, W.F. Remyelination of the superior cerebellar peduncle in old mice following demyelination induced by cuprizone. J Neurol Sci 1974, 22, 121-126, doi:10.1016/0022-510x(74)90059-8.
- Poliani, P.L.; Wang, Y.; Fontana, E.; Robinette, M.L.; Yamanishi, Y.; Gilfillan, S.; Colonna, M. TREM2 sustains microglial expansion during aging and response to demyelination. J Clin Invest 2015, 125, 2161-2170, doi:10.1172/JCI77983.
- Zhan, J.; Mann, T.; Joost, S.; Behrangi, N.; Frank, M.; Kipp, M. The Cuprizone Model: Dos and Do Nots. Cells 2020, 9, doi:10.3390/cells9040843.
- Ludwin, S.K. Chronic demyelination inhibits remyelination in the central nervous system. An analysis of contributing factors. Lab Invest 1980, 43, 382-387.
- Armstrong, R.C.; Le, T.Q.; Flint, N.C.; Vana, A.C.; Zhou, Y.X. Endogenous cell repair of chronic demyelination. Journal of neuropathology and experimental neurology 2006, 65, 245-256, doi:10.1097/01.jnen.0000205142.08716.7e.
- Harsan, L.A.; Steibel, J.; Zaremba, A.; Agin, A.; Sapin, R.; Poulet, P.; Guignard, B.; Parizel, N.; Grucker, D.; Boehm, N.; et al. Recovery from chronic demyelination by thyroid hormone therapy: myelinogenesis induction and assessment by diffusion tensor magnetic resonance imaging. J Neurosci 2008, 28, 14189-14201, doi:10.1523/JNEUROSCI.4453-08.2008.
- Safaiyan, S.; Besson-Girard, S.; Kaya, T.; Cantuti-Castelvetri, L.; Liu, L.; Ji, H.; Schifferer, M.; Gouna, G.; Usifo, F.; Kannaiyan, N.; et al. White matter aging drives microglial diversity. Neuron 2021, 109, 1100-1117 e1110, doi:10.1016/j.neuron.2021.01.027.
- Sen, M.K.; Mahns, D.A.; Coorssen, J.R.; Shortland, P.J. The roles of microglia and astrocytes in phagocytosis and myelination: Insights from the cuprizone model of multiple sclerosis. Glia 2022, 70, 1215-1250, doi:10.1002/glia.24148.
Round 3
Reviewer 2 Report
No comments.